# A biocompatible electrolyte enables highly reversible Zn anode for zinc ion battery

Guanjie Li [1], Zihan Zhao[1,2], Shilin Zhang [1] ✉, Liang Sun [1], Mingnan Li[1], Jodie A. Yuwono [1], Jianfeng Mao[1], Junnan Hao[1], Jitraporn (Pimm) Vongsvivut [3], Lidan Xing [4], Chun-Xia Zhao [1] & Zaiping Guo [1] ✉

Progress towards the integration of technology into living organisms requires power devices that are biocompatible and mechanically flexible. Aqueous zinc ion batteries that use hydrogel biomaterials as electrolytes have emerged as a potential solution that operates within biological constraints; however, most of these batteries feature inferior electrochemical properties. Here, we propose a biocompatible hydrogel electrolyte by utilising hyaluronic acid, which contains ample hydrophilic functional groups. The gel-based electrolyte offers excellent anti-corrosion ability for zinc anodes and regulates zinc nucleation/growth. Also, the gel electrolyte provides high battery performance, including a 99.71% Coulombic efficiency, over 5500 hours of long-term stability, improved cycle life of 250 hours under a high zinc utilization rate of 80%, and high biocompatibility. Importantly, the Zn//LiMn$_2$O$_4$ pouch cell exhibits 82% capacity retention after 1000 cycles at 3 C. This work presents a promising gel chemistry that controls zinc behaviour, offering great potential in biocompatible energy-related applications and beyond.

Soft wearable and implantable electronic devices that can seamlessly interface with the human skin hold immense potential for continuous and accurate monitoring of human health status and performance, with applications in various fields, such as military readiness, healthcare, and sports[1]. To support sensing, wireless communication, and signal conditioning in epidermal devices, power sources that primarily consist of energy storage systems, such as batteries are integrated[2]. However, conventional coin cells or thin-film batteries are bulky and rigid, making them unsuitable for skin interfacing. Although flexible energy-harvesting systems that use body motion or sweat may overcome this issue, they typically generate low and inconsistent energy output. Wireless harvesting of radiofrequency power through far- or near-field coupling with nearby antennas presents a promising approach, but it requires close proximity to a transmission antenna[3]. The lack of a universal solution for powering skin-interfaced devices has spurred research in advanced battery technologies, such as textiles

and flexible or stretchable systems, with a particular emphasis on lithium-ion or alkaline chemistries. However, their practicality and safety remain uncertain due to concerns regarding their long-term operation, reliance on toxic chemicals, and susceptibility to organic electrolyte leakage during operation. Therefore, batteries designed for bio-interfaced applications require advancements in soft and safe materials, robust battery chemistries, and biocompatible electrolytes to avoid safety hazards to the human body[4-7].

Rechargeable aqueous batteries have emerged as a promising option due to their use of inherently safe and inexpensive aqueous-based electrolytes. Among these systems, aqueous zinc metal batteries (AZMBs) are particularly attractive because of the utilisation of affordable and non-toxic Zn, which possess a favourable redox potential (−0.76 V vs. standard hydrogen electrode) and a high theoretical capacity (gravimetric capacity of 820 mAh g$^{-1}$ and volumetric capacity of 5855 mAh cm$^{-3}$)[8-11]. However, the Zn anode in AZMBs is

[1]School of Chemical Engineering, Faculty of Sciences, Engineering and Technology, The University of Adelaide, Adelaide, SA 5005, Australia. [2]Department of Dermatology of Shanghai Skin Disease Hospital, Institute of Psoriasis, Tongji University School of Medicine, Shanghai 200443, China. [3]Infrared Microspectroscopy (IRM) Beamline, ANSTO–Australian Synchrotron, 800 Blackburn Road, Clayton, VIC 3168, Australia. [4]School of Chemistry, South China Normal University, Guangzhou 510006, China. ✉e-mail: shilin.zhang01@adelaide.edu.au; zaiping.guo@adelaide.edu.au

plagued by irreversible issues, such as Zn dendrite growth during stripping/plating and parasitic side reactions in liquid electrolytes (e.g., hydrogen evolution reaction), which lead to continuous consumption of the Zn electrode and electrolyte[8,12–14]. These challenges pose practical barriers to the application of AZMBs, as they impair cycling stability and jeopardise the calendar life. To address these challenges, researchers have explored several strategies, including the development of aqueous/organic hybrid electrolytes, functional additives, and high-concentration electrolytes, to modify the $Zn^{2+}$ solvation structure and improve Zn reversibility[15–18]. However, these solutions may pose biosecurity concerns owing to their reliance on toxic chemicals. Furthermore, liquid electrolytes are prone to leakage when exposed to external stress in flexible AZMBs, and separators used in conjunction with liquid electrolytes can detach from the electrodes under strain. Hydrogel systems offer a promising alternative to liquid electrolytes owing to their superior flexibility and high fracture elongation. In particular, systems consisting of biocompatible and non-toxic polymers or bio-macromolecule assemblies exhibit intriguing potential in bio-interfaced applications. Hydrogel electrolytes, in particular, provide distinct advantages in suppressing parasitic water-relevant side reactions due to tailored hydrogen bonds between $H_2O$ and functional groups in hydrogels[19–23]. Despite these advantages, hydrogel electrolytes suffer from drawbacks related to the regulation of Zn deposition and unsatisfactory ionic conductivity, which restrict their ability to deliver long-lasting calendar life and high areal capacity with a deep depth of discharge (DoD) simultaneously[24–26].

Herein, we report an intrinsically biocompatible hydrogel electrolyte comprising hyaluronic acid (HA), water and $ZnSO_4$ salt, which displays exceptional performance when used in AZMBs. Symmetric cells assembled using this electrolyte demonstrate stable, long-term cycling for over 5500 h at a current density of 1 mA cm$^{-2}$ with a capacity of 1 mA h cm$^{-2}$, and over 250 h with a high Zn utilisation rate of 80% DoD$_{Zn}$, with an average Zn plating/stripping Coulombic efficiency (CE) of 99.71%. We confirm the biocompatibility of this hydrogel electrolyte using a WST-1 cell proliferation and cytotoxicity assay kit. More impressively, when paired with various cathodes, such as LiMn$_2$O$_4$ (LMO) and iodine (I$_2$), AZMBs with this hydrogel electrolyte exhibit an impressive ultra-long lifespan of over 1000 cycles for Zn//LMO batteries at a current rate of 3 C and over 20,000 cycles for Zn//I$_2$ batteries at 10 C, highlighting the potential of these flexible batteries for the use in wearable devices. Using correlated material characterisations, electrochemical examinations, in situ spectroscopic techniques, and theoretical computations, we elucidate that the enhanced performance of the hydrogel electrolyte is stemmed from the ability of the HA molecule to mitigate water reactivity and regulate the flux of $Zn^{2+}$/ $SO_4^{2-}$ ions. Our findings have significant practical implications for the development of high-performance hydrogel-based AZMBs for advanced soft wearable and implantable devices.

## Results

### Physicochemical properties of the HA gel electrolyte

The unique characteristics of water, such as its high permittivity and low viscosity, contribute to the fast ion migration and the intrinsic safety of aqueous electrolytes. However, water molecules also have detrimental effects, such as hydrogen (H$_2$) and oxygen (O$_2$) evolution, as well as anodic corrosion, ultimately leading to battery failure. To address these challenges, it is essential to minimise the concentration of water molecules at the electrode/electrolyte interface while maintaining the ion transfer capacity. This approach can effectively prevent $H_2O$-induced side reactions and improve the stability and efficiency of AZMBs[27–29].

The HA is a glycosaminoglycan consisting of diverse functional groups, e.g., −NH, −OH and −COO$^-$ (Supplementary Fig. S1), providing numerous bonding sites for $H_2O$ molecules. Therefore, HA has great potential to regulate the physical properties and coordination

structure of aqueous electrolytes. The incorporation of HA into the 2 M ZnSO$_4$ electrolyte leads to the formation of a gel state, which is different from the 2 M ZnSO$_4$ liquid electrolyte as illustrated in Fig. 1a. The gelation of HA−ZnSO$_4$−H$_2$O system is primarily attributed to the strong hydrogen bonding between HA and water, as evidenced by the formation of HA−H$_2$O gel in the absence of zinc salt (Supplementary Fig. S2). The gel formation is not limited to ZnSO$_4$ and can also occur in other zinc salt systems, such as zinc chloride and zinc trifluoromethyl sulfonate, which demonstrates a broader applicability of the HA in different zinc salt systems.

The measured ionic conductivity value for the HA gel electrolyte is slightly lower than that of the liquid electrolyte, decreasing from 57.0 to 47.7 mS cm$^{-1}$ (Supplementary Fig. S3), but it still outperforms other reported gel electrolytes (Supplementary Table S1) and is sufficient to provide rapid ion migration[30,31]. Supplementary Fig. S4a and b illustrate that the HA gel material displays a moderate elongation-at-break of 220% and a compressive strength of 0.18 MPa. The HA gel electrolyte can be stretched from 2 to 6 cm, representing a 200% increase in its original length, without any mechanical failure (Supplementary Fig. S4c). Additionally, by reducing the water content in the hydrogel, the mechanical strength of the HA gel electrolyte can be further enhanced for applications demanding high mechanical stability, at a minor cost of slightly decreased ionic conductivity (Supplementary Fig. S5). As summarised in Supplementary Table S2, the mechanical strength of the HA gel electrolyte is comparable to or even exceeds some of the previously reported gel electrolytes, indicating its suitability as a candidate for flexible applications.

To confirm the molecular structure of electrolytes, Fourier transform infra-red (FTIR) spectroscopy was utilised, and the characteristic stretching vibration related to HA and H$_2$O units can be observed in the FTIR spectra (Supplementary Fig. S6). The observed peaks at around 1556 cm$^{-1}$/1341 cm$^{-1}$ and 1417 cm$^{-1}$ correspond to the vibration of the −NH bond and −COO$^-$ bond in the HA molecule, respectively[32,33]. The chemical environment of $H_2O$ was further revealed by applying the curve-fitting approach to the observed FTIR spectra, as presented in Supplementary Fig. S7. The stretching vibration of the O−H in $H_2O$, which is located at 2900–3700 cm$^{-1}$, can be associated with three different types of H-bonds: strong (3205 cm$^{-1}$), weak (3410 cm$^{-1}$) and non-H-bond (3560 cm$^{-1}$)[34,35]. The fitting results in Supplementary Fig. S7c demonstrate that the HA gel electrolyte (87.2%) has a higher percentage of strong hydrogen bonds compared to that of the liquid electrolyte (81.5%). Additionally, it is noted that the presence of strong hydrogen bonds in HA−water hydrogel exhibits a noticeable increase, rising from 58.5% (in pure water) to 63.1% (HA−H$_2$O, Supplementary Fig. S8). These findings indicate that HA has a strong coordination capability with $H_2O$, which is further supported by Raman spectra shown in Supplementary Figs. S9 and S10. Molecular dynamics (MD) simulations (Fig. 1b) reveal that a single HA monomer can form 16.5 hydrogen bonds, which is much higher than the number of hydrogen bonds that can be formed by a single $H_2O$ molecule (1.53). Furthermore, density functional theory (DFT) calculations show the binding energy of H$_2$O−HA is more negative than that of H$_2$O−H$_2$O, indicating that HA forms stronger interactions with $H_2O$. The presence of HA in the gel electrolyte leads to a competition between HA and $H_2O$ molecules for hydrogen bonding, resulting in a reduction of hydrogen bonds between $H_2O$ and $H_2O$ by 6.9% (from 7200 to 6700, Supplementary Fig. S11). This decrease is consistent with the increase of strong hydrogen bonds by 5.7% (HA−H$_2$O binding) in the HA gel electrolyte observed by FTIR spectra (Supplementary Fig. S7), indicating a transformation of H$_2$O−H$_2$O bonds into HA−H$_2$O bonds. In addition, the electrochemical stability of both liquid and HA electrolytes was assessed by electrochemical stable windows (ESW), as demonstrated in Fig. 1c−e. Note that the platinum (Pt) was utilised as the working electrode for its significantly smaller HER overpotential compared to other high-overpotential electrodes such as titanium and glassy

carbon (Supplementary Table S3 and Fig. S12). It ensures that the HER occurs prior to the Zn deposition, allowing for the distinct identification and separation of the two processes during the linear sweep voltammetry test. The potentials corresponding to a current density of 0.1 mA cm$^{-2}$ are defined as the onset potentials for hydrogen evolution reaction (HER) and oxygen evolution reaction (OER). Due to the highly interconnected hydrogen bonds network in the HA gel electrolyte, the reactivity of $H_2O$ is effectively suppressed, resulting in a lower potential for HER (−0.52 vs. −0.36 V, Fig. 1c). Concurrently, the potential of the OER shows a slight increase, accompanied by a reduced current response (Fig. 1d). These findings suggest that the ESW can be extended in our HA gel electrolyte, and the water splitting on both cathode and anode electrodes can be suppressed as well[36].

### Anti-corrosion ability of Zn anodes in the HA gel electrolyte

Anti-corrosion ability of Zn anodes in the HA gel electrolyte is dependent on the number of free protons in the electrolyte[37–40]. X-ray diffraction (XRD), scanning electron microscopy (SEM) images, and electrochemical examinations were employed to investigate the Zn metal corrosion. As shown in Fig. 2a, the addition of HA to the liquid electrolyte increases the pH value from 4.25 to 5.05, indicating that HA can effectively reduce the number of free $H^+$ ions and prevent anodic corrosion compared to the liquid electrolyte. Furthermore, the pH value of 6.93 in 1 wt% HA and pure water confirm that HA is a neutral molecule in water, unlike traditional pH buffers such as $CH_3COOH$ that rely on their own protons to adjust the pH value of electrolytes[41].

Instead, the pH increases in the HA gel electrolyte are ascribed to the immobilisation of dissociative $H^+$ by the O- and N-containing functional groups of HA through strong H-bonds formation. To estimate the corrosion rate, the corrosive current was measured by using the Tafel test. Figure 2b indicates that the Zn anode in the HA gel electrolyte exhibits a lower corrosion current of 0.297 mA cm$^{-2}$ compared to that in the liquid electrolyte (1.632 mA cm$^{-2}$). To accurately quantify the corrosion rate of the Zn anode, we monitor the potential variation of a Zn@Ti electrode, where a predetermined amount of Zn was plated onto a Ti foil. As is shown in Supplementary Fig. S13, the potential of the Zn@Ti electrode using the liquid electrolyte demonstrates a sudden increase only after 38 h, indicating a complete consumption of the metallic Zn due to corrosion reactions[42]. The corrosion rate (Corr) can be obtained by the equation: Corr $= \frac{m_{Zn}}{t}$ (Eq. (1)), where $m_{Zn}$ is the total mass of deposited Zn metal and $t$ is the corrosion time corresponding to the complete consumption of Zn. Accordingly, the corrosion rate was calculated to be 0.0363 mg h$^{-1}$, which is significantly higher than the corrosion rate of 0.0068 mg h$^{-1}$ observed with the HA gel electrolyte. This obvious difference in corrosion rates provides clear evidence of the excellent anti-corrosion capability of the HA gel electrolyte. An immersion experiment was further conducted to investigate the corrosion behaviour of Zn metals in two electrolytes. Upon immersion of Zn metals in the liquid electrolyte, the pH value of the electrolyte rapidly increases from 4.25 to 5.20 within 3 days, indicating the corrosive nature of the electrolyte (Supplementary Fig. S14). The pH value was continuously increased to 5.45 after soaking for

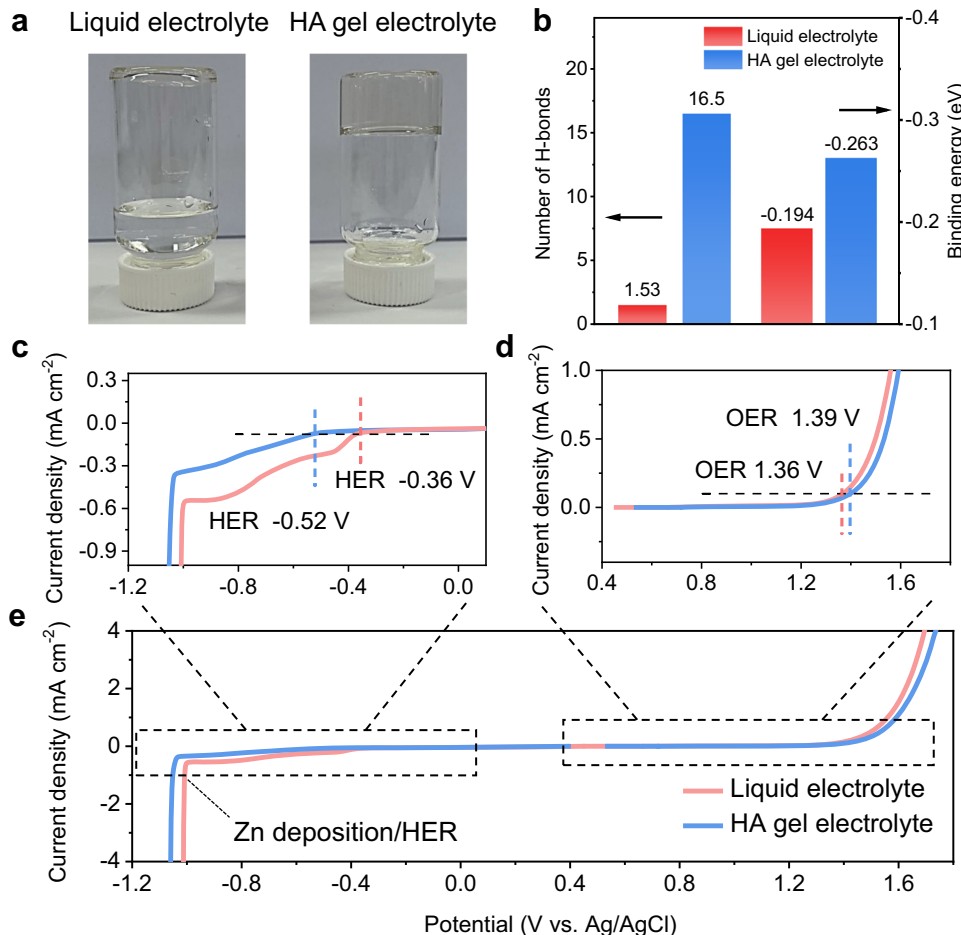

**Fig. 1 | Characterisation of physical properties and electrochemical stable window of the HA gel electrolyte. a** Optical photos of liquid and HA gel electrolytes. Liquid electrolyte: 2 M $ZnSO_4$ aqueous solution. **b** Calculated number of H-bond of single $H_2O$ and HA molecule and binding energies of $H_2O$–$H_2O$ and $H_2O$–HA. Magnified views of the regions outlined near **c** anodic and **d** cathodic extremes in **e** overall electrochemical stability window of liquid and HA gel electrolytes on non-active Pt electrodes. HER hydrogen evolution reaction, OER oxygen evolution reaction.

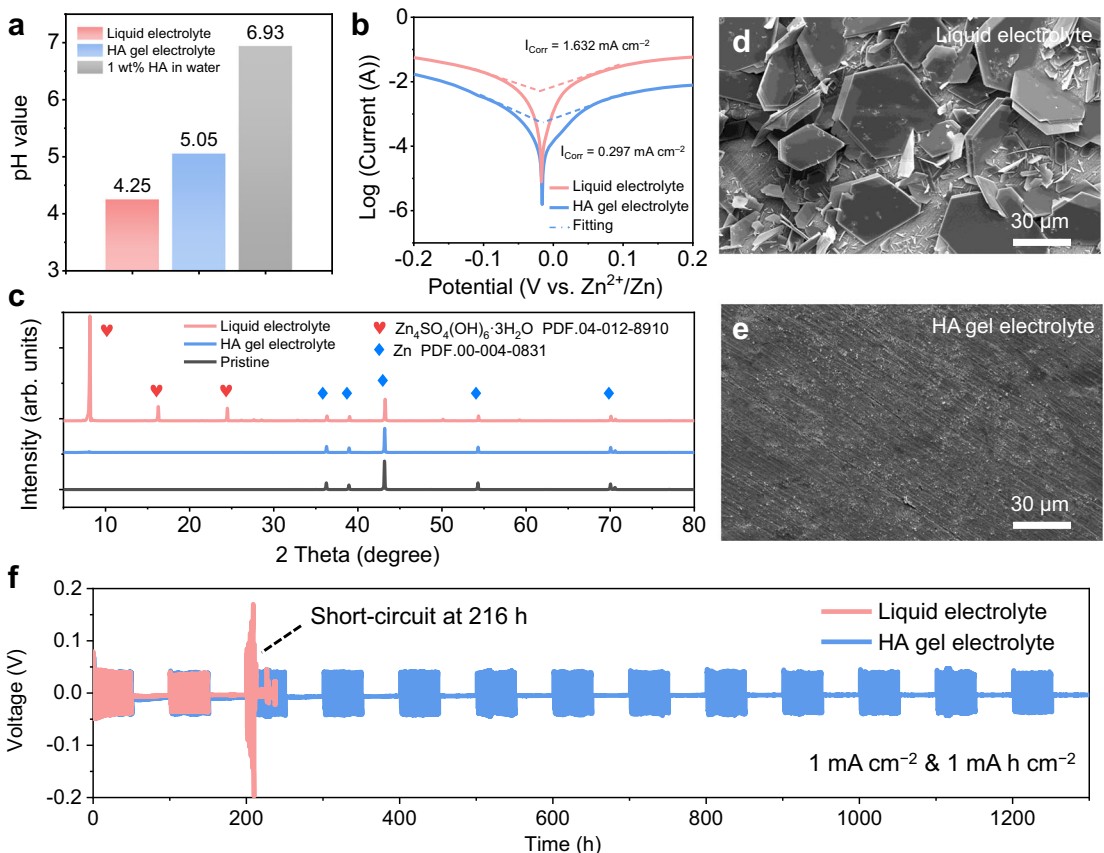

**Fig. 2 | Characterisation of Zn anodic corrosion behaviour. a** Measured pH values of the liquid and HA gel electrolytes. **b** Tafel plots of Zn anode in both liquid and HA gel electrolytes, showing corrosive currents ($i_{corr}$) and potentials. **c** XRD pattern of the Zn metal before and after immersing in both liquid and HA gel electrolytes for 12 days. PDF stands for powder diffraction file. SEM images of soaked Zn metal in: **d** the liquid electrolyte and **e** the HA gel electrolyte. **f** Cycle performance of Zn//Zn symmetric cells during intermittent galvanostatic charge/discharge test at a current density of 1 mA cm$^{-2}$, with a capacity of 1 mA h cm$^{-2}$. The intermittent galvanostatic charge/discharge test allows the corrosion reaction to occur at a certain charge/discharge interval, simulating the actual corrosion behaviours under practical working conditions.

12 days, resulting in the formation of a significant amount of $Zn_4SO_4(OH)_6 \cdot 3H_2O$ (ZHS) by-product with a hexagonal plate structure, which deposited on the surface of the Zn electrode (Fig. 2c, d and Supplementary Fig. S15). The formation of ZHS by-product results from the complexation reaction between $Zn^{2+}/SO_4^{2-}$ and $OH^-$ that remains in the electrolyte during the corrosion of the Zn anode by $H^+$ and the subsequent release of $H_2$ from the electrolyte[43–46]. In contrast, negligible pH variation and fewer by-products were observed when Zn was immersed in the HA gel electrolyte, indicating an exceptional corrosion tolerance capability (Fig. 2e and Supplementary Fig. S15). To further understand this phenomenon, the dynamic change in interfacial impedance of Zn//Zn symmetric cells during the resting process of AZMBs was examined. Nyquist plots and corresponding fitting equivalent circuits are presented in Supplementary Fig. S16. As shown in Supplementary Fig. S17 and Table S4, after soaking the Zn metal in liquid electrolyte for 7 days, the resistance of bulk electrolyte ($R_e$) significantly increased from 3.8 to 15.7 Ω, due to continuous corrosion between the Zn metal and the electrolyte, which consumes the electrolyte and hinders ionic transportation. The ZHS by-products formed on the Zn surface have a porous structure that cannot prevent continuous Zn corrosion during immersion. Furthermore, these ZHS by-products have poor ionic conductivity that significantly impedes the charge transfer, leading to a sharp increase in both SEI film resistance ($R_f$) and charge transfer resistance ($R_{ct}$) in Zn//Zn symmetric cells. As a result, the $R_{ct}$ in the liquid electrolyte can reach up to 7740 Ω after 7 days of immersion. This increase in interfacial impedance ultimately leads to the compromised longevity of the Zn anode, as confirmed by

the shortened time to reach a short circuit as the storage time increases (Supplementary Fig. S18). Additionally, SEM images reveal that the Zn deposition is mostly inhibited on the surface of Zn covered by ZHS products (Supplementary Fig. S19), leading to uneven charge distributions and the growth of Zn dendrites, which is the primary mechanism for the failure of Zn anodes induced by ZHS. In contrast, the cycling performance results demonstrate that the cycle life of Zn//Zn cells remains stable even after aging in the HA gel electrolytes for 12 days, despite a slightly increased polarisation in the initial cycle (Supplementary Fig. S20a). The symmetric Zn//Zn cells can operate for up to 300 h without an increase in electrode polarisation (Supplementary Fig. S20b). Additionally, as demonstrated by SEM images in Supplementary Fig. S21, there is no ZHS deposited on the immersed Zn anode, even after the 12-day aging process in the HA gel electrolyte. These findings provide further evidence of the effectiveness and durability of the HA hydrogel electrolyte in maintaining a stable chemical environment and enhancing the performance of the Zn anode. Therefore, the symmetric Zn//Zn cells using the HA gel electrolyte demonstrate an ultra-stable cycle life of over 1200 h during the intermittent charge/discharge test, which is much longer than the 216 h for Zn/Zn symmetric cells using the liquid electrolyte (Supplementary Fig. 2f).

### Regulation of Zn nucleation/growth in the HA gel electrolyte
Dendrite formation during charge–discharge poses a significant challenge for AZMBs, which is closely related to the behaviour of Zn nucleation and growth. To gain insights into the Zn nucleus radius and

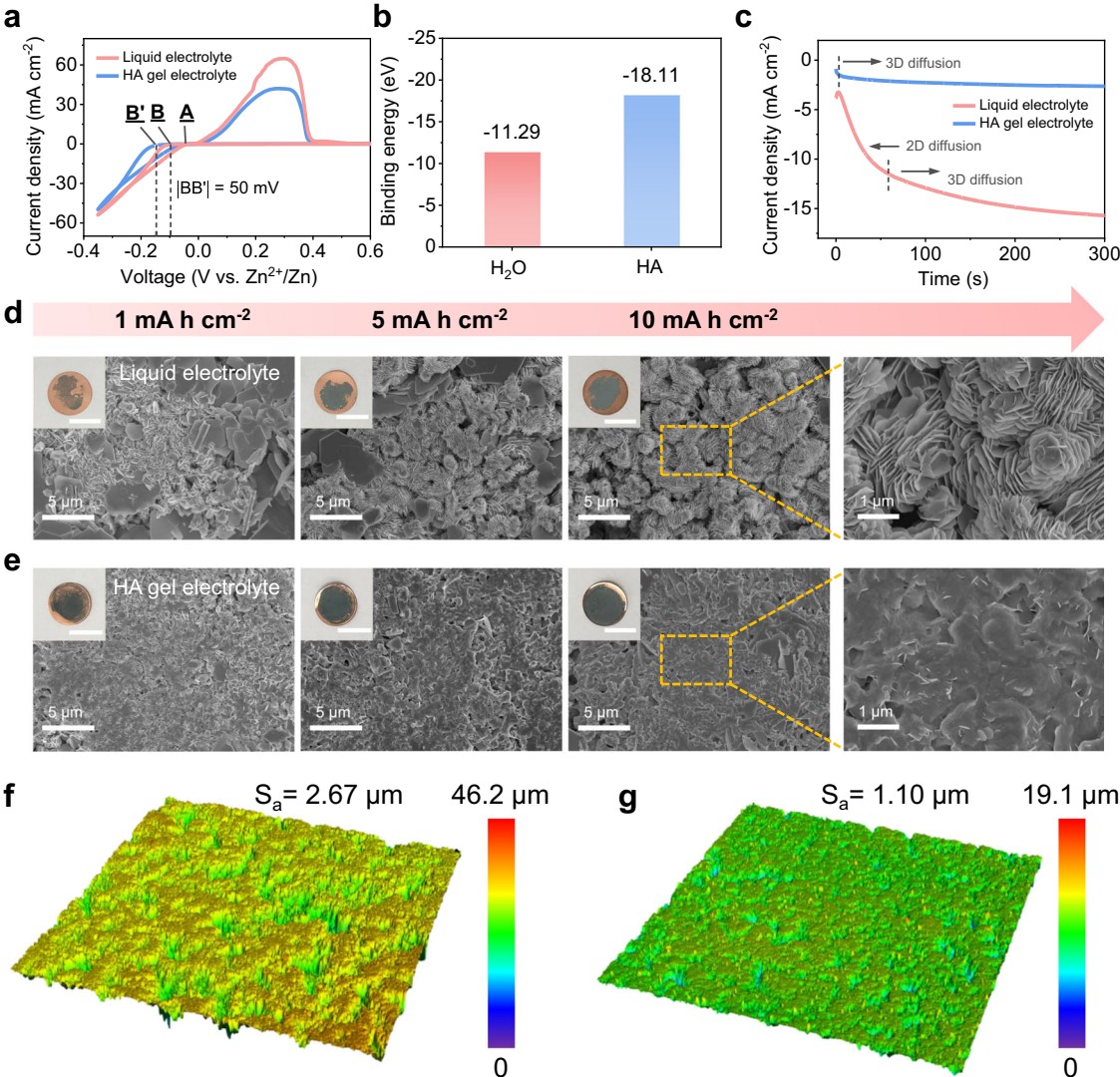

**Fig. 3 | Nucleation and growth behaviours of Zn anode. a** Cyclic voltammograms of Zn//Cu cells in both liquid and HA gel electrolytes at a scanning rate of 5 mV s⁻¹. Point A represents the crossover potential, point B and B' the onset potential of Zn²⁺ ions deposition reactions. **b** Calculated binding energies of Zn²⁺–H₂O and Zn²⁺–HA. **c** Chronoamperogram test of Zn//Zn symmetric cells in both liquid and HA gel electrolytes at a constant voltage of −150 mV. SEM images of plating Zn on Cu current collectors at a constant current density of 1 mA cm⁻² with different areal capacities in **d** the liquid electrolyte and **e** the HA gel electrolyte. Insets show optical photos of the corresponding Zn@Cu anodes. Scalebar: 1 cm. Three-dimensional confocal laser microscopy images of Zn plating on Cu current collectors with a capacity of 10 mAh cm⁻² in **f** the liquid electrolyte and **g** the HA gel electrolyte. Sa arithmetic mean height.

density, cyclic voltammetry tests were performed on Zn//Cu asymmetric cells to evaluate the nucleation overpotential (NOP). Figure 3a shows that the NOP of the Zn deposition on Cu foil in the HA gel electrolyte is 50 mV higher than that of Zn anode in the liquid electrolyte. This suggests that a smaller nucleus radius is formed when the Zn anode cycles in the HA gel electrolyte, resulting in the formation of fine-grained and uniform Zn particles during Zn deposition[47,48]. The large NOP is also supported by the electrochemical plating behaviour of Zn on the Cu foil at a current density of 1 mA cm⁻² (Supplementary Fig. S22). The increased overpotential is attributed to the large binding energy between the Zn²⁺ ion and HA molecule (Fig. 3b), although at the expense of slightly sacrificing ionic conductivity. To further investigate the growth mechanism of Zn nuclei, chronoamperometry (CA) measurements were conducted. As shown in Fig. 3c, the Zn anode in the liquid electrolyte exhibits a rapid 2D diffusion process with a fast current increase (absolute value) lasting over 60 s, indicating a rapid expansion of the specific surface area due to the uncontrolled growth of a porous Zn deposition layer[49–51]. In contrast, the HA gel electrolyte suppresses the 2D diffusion process

and directly reaches a constant 3D diffusion process with a steady and low current, suggesting that the Zn nuclei slowly evolve to form a uniform Zn layer. This result is in accordance with the morphological evolution of Zn deposition in SEM and optical images (Fig. 3d, e and Supplementary Fig. S23). As the deposition capacity increased from 1 mAh cm⁻² to 10 mAh cm⁻², Zn flakes tend to grow vertically in the liquid electrolyte, resulting in loosely stacked Zn clusters (Fig. 3d), which is consistent with the typical 2D diffusion process inferred from the CA results (Fig. 3c). In contrast, in the HA gel electrolyte, Zn flakes stack parallel to each other and grow horizontally, gradually forming an evenly distributed and compact structure (Fig. 3e). The thickness of this compact Zn layer in the HA gel electrolyte is only 24 μm, which is much thinner than that in the liquid electrolyte (32 μm, Supplementary Fig. S24).

Importantly, the HA is typically present in the form of a sodium salt that dissociates into HA anions and Na⁺ cations in the electrolyte. The negatively charged HA can act as an anion surfactant to shield overcharged sites and homogenise the local current distribution. Importantly, unlike conventional anion surfactants, the large skeleton

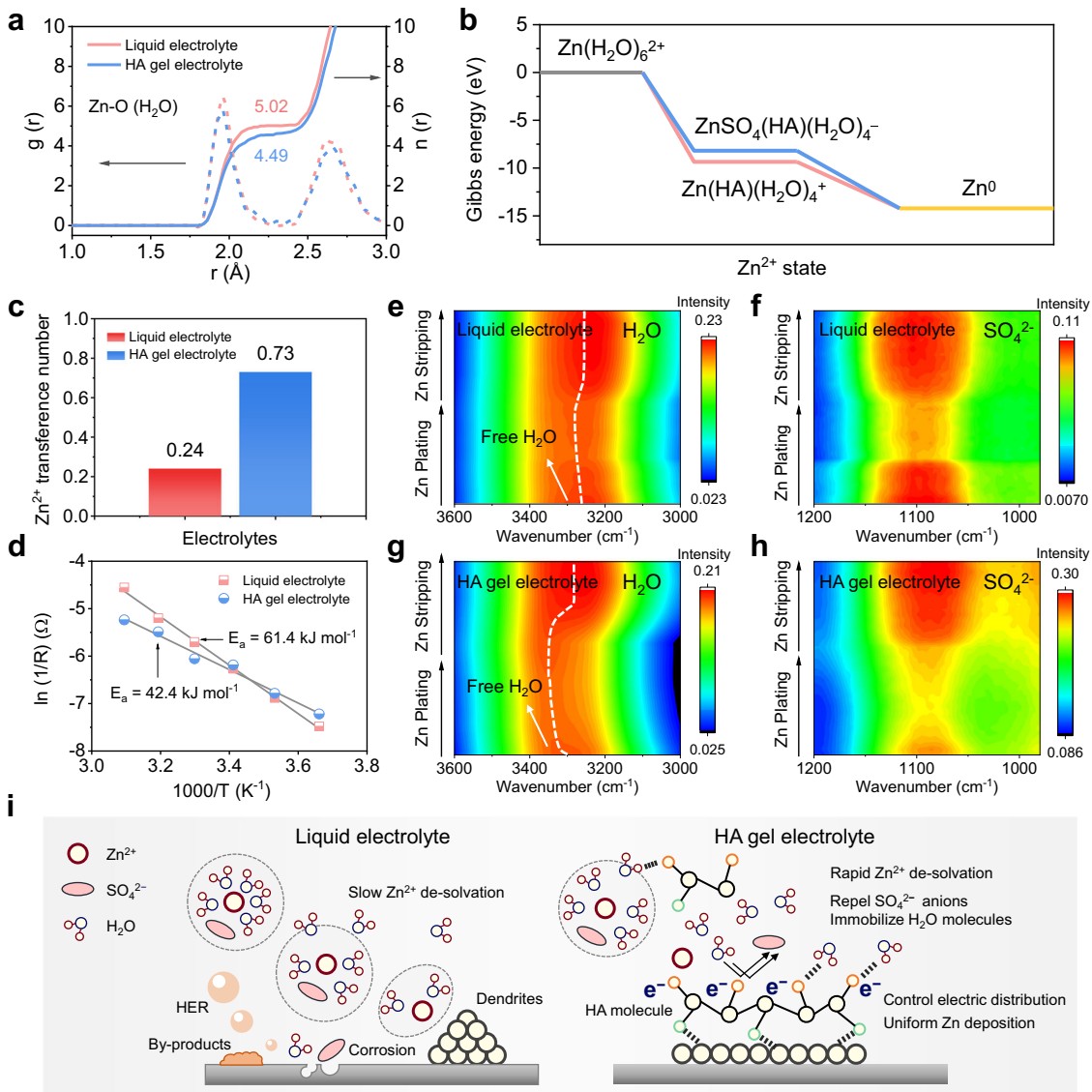

**Fig. 4 | Mechanism understanding of the regulation of Zn electrodeposition.**
**a** The radial distribution function $g(r)$ and coordination number $n(r)$ of $Zn^{2+}$–O ($H_2O$) obtained from MD simulations in both liquid and HA gel electrolytes. **b** Gibbs free energy of different $Zn^{2+}$ solvation structures. **c** Transference number of $Zn^{2+}$ in both liquid and HA gel electrolytes. **d** Arrhenius curves and comparison of activation energies ($E_a$) for de-solvation of hydrated $Zn^{2+}$ ions in both liquid and HA gel electrolytes. In-situ synchrotron-FTIR spectra in attenuated total reflection (ATR) mode of stainless-steel electrode during Zn plating/stripping: **e** $\nu$ (OH)–liquid electrolyte, **f** $\nu$ ($SO_4^{2-}$)–liquid electrolyte, **g** $\nu$ (OH)–HA gel electrolyte, **h** $\nu$ ($SO_4^{2-}$)–HA gel electrolyte. **i** Schematic illustration of the regulation mechanism of HA gel electrolyte on Zn electroplating.

size and gelation state of HA prevent it from being repelled from the negatively charged Zn surface, resulting in a long-lasting "electric field shielding effect"[52,53]. This unique property enables the HA gel electrolyte to achieve almost complete coverage of the Cu collector with uniform Zn electroplating, even at a low capacity of 1 mAh cm$^{-1}$ (Fig. 3e, optical images in the upper left corner), indicating the formation of uniform Zn electroplating. To visualise the topological morphology and roughness of the Zn deposition layer, the confocal laser microscope was used. Figure 3f, g reveal a significant difference in surface roughness (arithmetic mean height, Sa) between the Zn layer in the liquid electrolyte, which exhibits a large Sa of 2.67 μm, and that in the HA gel electrolyte, which has a much lower Sa of only 1.10 μm. The corresponding 2D mappings also indicate that Zn particles are uniformly distributed in the HA gel electrolyte, as shown in Supplementary Fig. S25. These observations provide strong evidence for the good regulation capabilities of the HA gel electrolyte in controlling both Zn nucleation and growth.

The mechanism underlying the impact of the HA gel electrolyte on Zn deposition was further investigated at the molecular level. Typically, the de-solvation process is the rate-limiting step for Zn deposition due to the strong binding between $Zn^{2+}$ and $H_2O$, which requires an overwhelming amount of de-solvation energy[54–57]. A delay or inefficiency in the de-solvation process can disrupt the $Zn^{2+}$ flux, leading to the accumulation of $H_2O$ or $SO_4^{2-}$ at the electrolyte/electrode interface, causing irregular Zn deposition behaviour and anodic corrosion[58–60]. To gain a better understanding of the effect of HA on the solvation structure of $Zn^{2+}$, MD simulations were performed. Radial distribution functions (RDFs) in Fig. 4a show that the average coordination number (ACN) of $H_2O$ molecules surrounding $Zn^{2+}$ is 5.02 in the 2 M $ZnSO_4$ liquid electrolyte. Noted that the ACN 5.02 in the liquid electrolyte means that the solvation structures are made from the combination of dominated cluster of $Zn(H_2O)_6^{2+}$ and other clusters such as $ZnSO_4(H_2O)_5$ and $ZnSO_4(H_2O)_4$ (Supplementary Figs. S26–S28, simulations data are provided as Supplementary Data 1 file). The ACN

of $H_2O$ in the solvation shell decreases to 4.49 in the HA gel electrolyte due to the strong bonding between $H_2O$ and HA, which reduces the electrostatic interactions of $H_2O$ with $Zn^{2+}$. In addition, despite the steric hindrance and large molecular structure of HA, a small number of HA molecules can penetrate the inner solvation structure of $Zn^{2+}$ due to the high binding energy between HA and $Zn^{2+}$ (Supplementary Fig. S29). According to the DFT calculation, the Gibbs free energy of $Zn(HA)(H_2O)_4^+$ is significantly lower than that of water-dominated $Zn(H_2O)_6^{2+}$ (Fig. 4b), indicating that the presence of $Zn(HA)(H_2O)_4^+$ is thermodynamics favourable. This finding provides additional evidence to support the formation of the $Zn^{2+}$-HA solvation structure. Moreover, the negatively charged HA skeleton can repel the $SO_4^{2-}$ from the $Zn^{2+}$ solvation structure and weaken the coordination between $SO_4^{2-}$ and $Zn^{2+}$. This is supported by the lower Gibbs free energy of $Zn(HA)(H_2O)_4^+$ compared to $ZnSO_4(HA)(H_2O)_4^-$ (Fig. 4b). Besides, the ion concentration and distribution near the Zn electrode were determined by measuring the capacitance of electric double layer (EDL) on the Zn electrode. The EDL capacitance ($C_{EDL}$) is governed by the equation $C_{EDL} = \frac{\varepsilon A}{d}$ (Eq. (2)), where $\varepsilon$ is the electrolyte dielectric constant, $A$ is electrode surface area and $d$ is the thickness of EDL. Since both the liquid electrolyte and the HA gel electrolyte use water as the solvent and have the same Zn electrode size, $\varepsilon$ and $A$ can be considered roughly identical for comparison purposes. As a result, the $C_{EDL}$ is mainly influenced by the thickness of the EDL, exhibiting a negative correlation. As shown in Supplementary Fig. S30, the EDL capacitance increases from 111 to 177 $\mu F\, cm^{-1}$ in the HA gel electrolytes, indicating a smaller $d$ value in the presence of the HA gel electrolyte. Based on the classic stern model[61], this observation verifies that a greater concentration of $Zn^{2+}$ ions but a reduced concentration of $SO_4^{2-}$ ions accumulate on the surface of the Zn electrode in the HA gel electrolyte, attributed to the weak coordination between $Zn^{2+}$ and $SO_4^{2-}$ ions (Supplementary Fig. S31). The red shift of $SO_4^{2-}$ in FTIR spectra in Supplementary Fig. S32 also verifies the weak chemical environment[62]. This unique chemical coordination enables the oriented migration of electrolyte species, which contributes to a higher $Zn^{2+}$ transference number in the HA gel electrolyte (0.73) compared with that in the liquid electrolyte (0.24, Fig. 4c and Supplementary Fig. S33). The high $Zn^{2+}$ transference number plays a crucial role in enhancing the electrochemical performance of the zinc anode through two key mechanisms: (1) allows for efficient transport of $Zn^{2+}$ ions, alleviating the concentration gradient and ensuring a uniform $Zn^{2+}$ ion distribution and thereby the dendrite-suppressed Zn deposition. (2) decreases anion concentration near the Zn anode and mitigates $SO_4^{2-}$-induced corrosion reactions[63,64].

As discussed, HA can weaken the electrostatic interactions between $Zn^{2+}$ and $H_2O$ or $SO_4^{2-}$, which helps to facilitate the de-solvation process of $Zn^{2+}$ near the electrode/electrolyte interface. As expected, the de-solvation energy in the HA gel electrolyte is significantly reduced to 42.4 $kJ\, mol^{-1}$, which is much lower than 61.4 $kJ\, mol^{-1}$ for the liquid electrolyte (Fig. 4d and Supplementary Fig. S34). Afterward, in-situ synchrotron-FTIR tests were performed to monitor dynamic changes in chemical species at the electrode/electrolyte interface during Zn deposition-dissolution. As shown in Fig. 4e–h, both the $H_2O$ molecule and $SO_4^{2-}$ anion present a reversible change in two electrolytes, which is in good accordance with the plating-stripping process of the Zn anode. During the Zn plating process, $H_2O$ and $SO_4^{2-}$ are expelled from the interface due to the de-solvation of $Zn^{2+}$, leading to an increase in free water (indicated by a blueshift in the O−H bond) and a decrease in $SO_4^{2-}$ ions (shown as intensity attenuation). The opposite occurs during the Zn stripping process[65,66]. Notably, the HA gel electrolyte demonstrates a noticeable blueshift in the O-H bond towards shorter wavelengths and a decrease in the intensity of $SO_4^{2-}$ ions, as shown in Fig. 4g, h. This confirms the rapid removal of both $H_2O$ and $SO_4^{2-}$ from the solvation shell of $Zn^{2+}$ during Zn deposition. This, in turn, facilitates the homogenisation of

$Zn^{2+}$ flux and uniform nucleation/growth of Zn during plating/stripping[67,68]. Obviously, our HA gel electrolyte plays a crucial role in regulating Zn growth and deposition because (1) it weakens the electrostatic interactions between $Zn^{2+}$ and $H_2O/SO_4^{2-}$ by means of its strong binding energy with $H_2O$ and negatively charged groups, promoting the faster $Zn^{2+}$ de-solvation and reducing the residual $H_2O/SO_4^{2-}$ on the Zn surface due to the incomplete $Zn^{2+}$ de-solvation; (2) it reduces the high surface energy on the tips of the protrusions via the electrostatic shielding effect, leading to a more uniform local current distribution and therefore the homogeneous Zn deposition (Fig. 4i). These findings highlight the importance of our HA gel electrolyte in achieving uniform and controlled Zn growth and deposition, which has significant implications for the development of high-performance Zn-based batteries.

## Long-term electrochemical stability of Zn anodes

The HA gel electrolyte used in this study is anticipated to provide superior electrochemical performance due to its unique properties. Figure 5a exhibits the long-term cycle stability of Zn//Zn symmetric cells with a capacity of 1 mAh $cm^{-2}$ at a current density of 1 mA $cm^{-2}$. The symmetric cell using the HA gel electrolyte shows an ultra-long lifespan of 5500 h with a stable overpotential of around 50 mV. In contrast, cells using the liquid electrolyte fail after only 482 h, possibly due to the dendrite-induced short circuits. In addition, despite the lower ionic conductivity of the HA gel electrolyte, the rate capability of Zn//Zn symmetric cells using the HA gel electrolyte is comparable to that using the liquid electrolyte (Supplementary Fig. S35). At a current density of 5 mA $cm^{-2}$ and a capacity of 5 mAh $cm^{-2}$, the HA gel electrolyte exhibits an extended cycling life of up to 1200 h (Supplementary Fig. S36). These findings demonstrate that the HA gel electrolyte can facilitate the stable and efficient operation of Zn-based batteries.

Additionally, post-mortem examinations were conducted to study the underlying mechanism responsible for the improved stability of the Zn anode in the HA gel electrolyte (Fig. 5b–e). After 100 continuous plating/stripping cycles, the Zn surface in the liquid electrolyte exhibits a rough texture with agglomerating Zn particles, forming a thick deposition layer with a thickness of 54 $\mu m$ (Fig. 5b, d). In contrast, a smooth and compact deposition layer with a thickness of 16 $\mu m$ is observed in the Zn electrode cycled in the HA gel electrolyte (Fig. 5c, e). This finding is consistent with the reconstructed 3D morphologies presented in Supplementary Fig. S37, where the Zn in the HA gel electrolyte exhibits a much lower Sa value of 2.90 $\mu m$ compared to that in the liquid electrolyte (13.05 $\mu m$). Optical images further confirm the uniform deposition of Zn in the HA electrolyte (Supplementary Fig. S38). It is worth noting that the white colour of plating Zn in the liquid electrolyte differs from that of pristine Zn or cycled Zn in the HA gel electrolyte. The white substance was identified as the ZHS by-products resulting from the corrosion reaction, as indicated by the XRD result in Supplementary Fig. S39. This is because the parasitic HER during charge/discharge further aggravates the increase of the localised concentration of $OH^-$, exacerbating the corrosion process and producing a considerable amount of ZHS[69,70]. In contrast, the HA gel electrolyte suppresses the HER reaction and related corrosion reactions due to the reduced activity of water and improved electrochemical stability. As a result, the Zn plating/stripping efficiency of the Zn//Cu cell is significantly improved in the HA gel electrolyte, with an impressive average Coulombic efficiency of 99.71% after 2000 cycles at a current density of 1 mA $cm^{-2}$, with a capacity of 1 mAh $cm^{-2}$ (Fig. 5f and Supplementary Fig. S40). Compared with recently reported gel electrolytes[7,9,10,21–23,25,26,39,46,56,57], the HA gel electrolyte demonstrates superior performance in both Coulombic efficiency (99.71%) and cumulative capacity (3 Ah $cm^{-2}$) for the Zn anode (Fig. 5g and Supplementary Table S5). To evaluate the potential of the HA gel electrolyte for practical applications, the cycle performance of

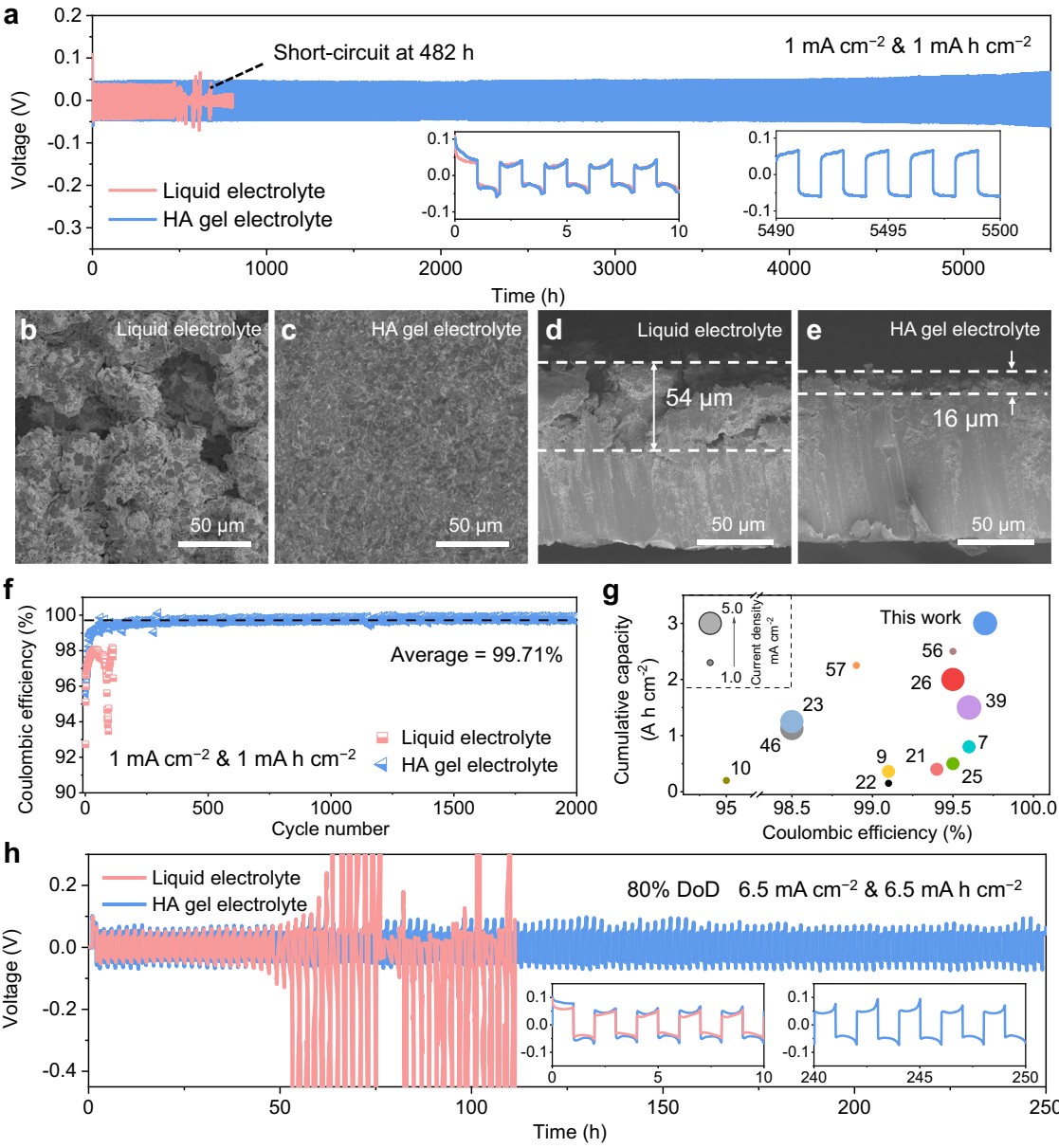

**Fig. 5 | Electrochemical characterisation of Zn anode. a** Long-term cycle performance of Zn//Zn symmetric cells in both liquid and HA gel electrolytes at a current density of 1 mA cm$^{-2}$ with a capacity of 1 mAh cm$^{-2}$. Top and cross-section SEM images of Zn metal after 100 cycles in the liquid electrolyte (**b**, **d**) and the HA gel electrolyte (**c**, **e**). **f** Long-term cycle performance of Zn//Cu asymmetric cells in both liquid and HA gel electrolytes at a current density of 1 mA cm$^{-2}$ with a capacity of 1 mAh cm$^{-2}$ at an upper cut-off voltage of 0.5 V. **g** The comparison of cumulative capacities and average Coulombic efficiency with previously reported gel electrolytes. The number refers to the references. **h** Cycle performance of Zn//Zn symmetric cells with a high Zn utilisation rate of 80% depth of discharge (DoD$_{Zn}$).

the Zn anode was further investigated under a medium DoD$_{Zn}$ of 33% and a high DoD$_{Zn}$ of 80%. Remarkably, in the HA gel electrolyte, the Zn//Zn symmetric cell delivers a stable cycling performance for 650 h under 33% DoD$_{Zn}$ and even 250 h under 80% DoD$_{Zn}$ (Fig. 5h and Supplementary Fig. S41). This performance is comparable to that of state-of-the-art cells employing gel electrolytes (Supplementary Table S6). This impressive performance demonstrates that our HA gel electrolyte holds promise for practical applications where thin Zn foils (<10 μm) are necessary to enhance the volumetric energy density and reduce the size of AZMBs for wearable or implanted devices.

### Biocompatibility and applications of the HA gel electrolyte
The applicability of hydrogel electrolytes for use in soft wearable and implantable devices largely depends on the biocompatibility of these materials. Therefore, we evaluated the biocompatibility of the

hyaluronic acid gel electrolyte using the MST-1 cell proliferation assay kit. Primary epidermal keratinocytes (HEKn) isolated from the skin of male neonates were utilised to mimic the interaction between the HA gel electrolyte and human skin cells. Additionally, RAW 264.7 macrophage cells lined from the tumour of male mice were employed to explore the effect of HA on cells adjacent to the implanted area, as these cells play a critical role in wound healing, apoptotic cell clearance, and tissue re-modelling[6,71]. HEKn and RAW 264.7 cells were cultured in the medium with or without the HA gel electrolyte for 24 and 48 h, and the cell viability was measured using fluorescent plate readers. As shown in Fig. 6a, b, the HA gel electrolyte has a negligible impact (not significant in statistics) on the viability of both HEKn and RAW 264.7, demonstrating minimal cellular cytotoxicity of the HA gel electrolyte. This excellent biocompatibility ensures the promising application of the as-prepared hydrogel in wearable and implanting devices.

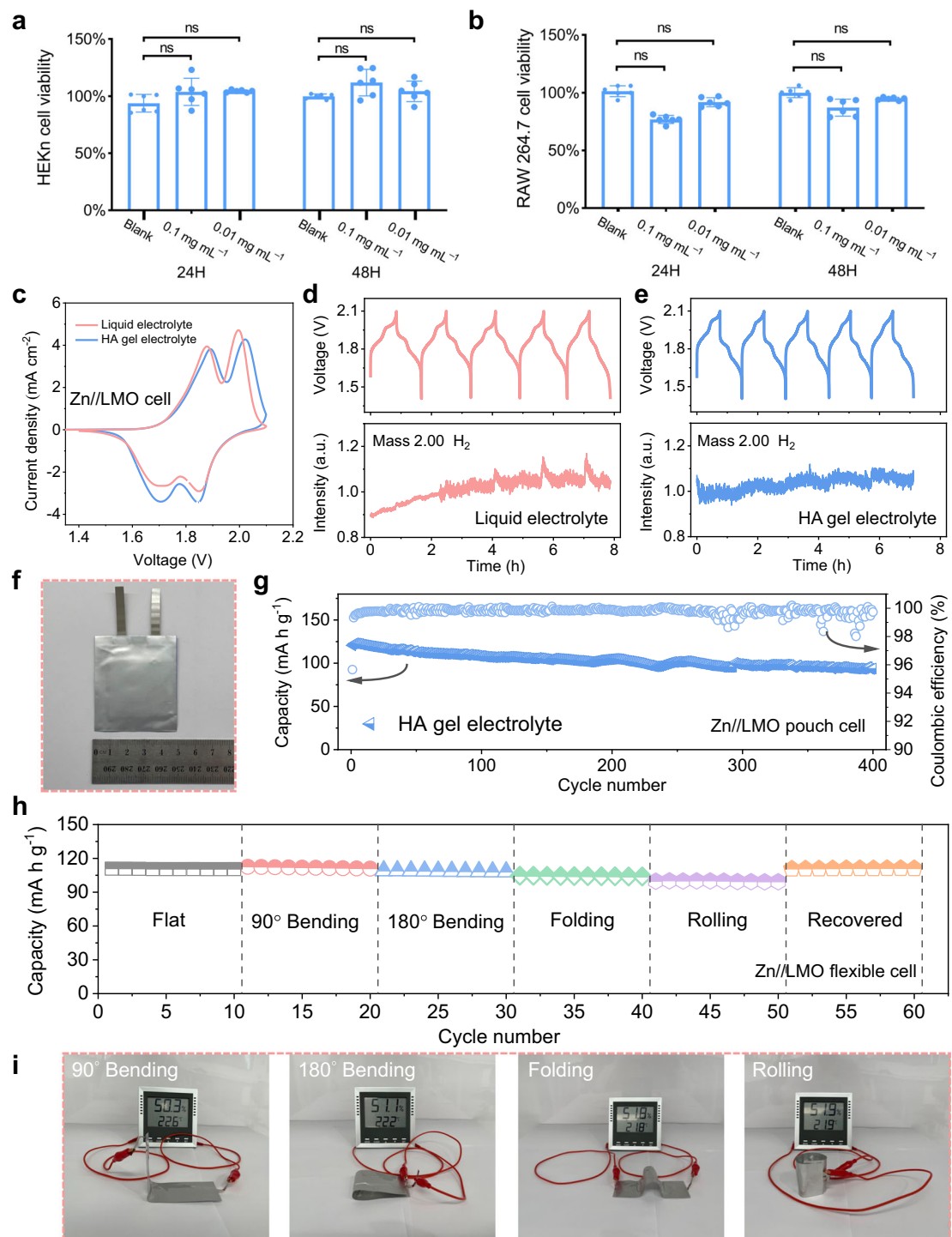

**Fig. 6 | Biocompatibility and flexible batteries applications.** Cell viability of **a** HEKn and **b** RAW 264.7 cells incubated in the medium containing HA electrolytes. The error bars represent the standard deviation of the values, ns stands for not significant in statistics. **c** Cyclic voltammograms of Zn//LMO cells in both liquid and HA gel electrolytes with a voltage range of 1.4–2.1 V at a scanning rate of 1 mV s⁻¹. Differential electrochemical mass spectrometry spectra of Zn//LMO cells upon charge/discharge at a current density of 1 C in **d** the liquid electrolyte and **e** the HA gel electrolyte. **f** Optical photo of the assembled Zn//LMO pouch cell. **g** Long-term cycle performance of the Zn//LMO pouch cell using the HA gel electrolyte at a current rate of 1 C. **h** Cycle performance of flexible Zn//LMO battery using the HA gel electrolyte under various mechanical deformation status. **i** Optical photos of a hygrometer powered by the flexible Zn//LMO battery under varied deformation conditions.

The biocompatible HA gel electrolyte was further assembled into high-voltage Zn//LiMn₂O₄ (LMO) hybrid AZMBs to investigate its potential for practical applications. The mass loading of LMO cathodes was about 11 mg cm⁻², capable of providing an areal capacity of about 1.63 mAh cm⁻². The coin-typed cell structure using the hydrogel electrolyte is illustrated in Supplementary Fig. S42. Despite a slight increase in electrode polarisation, the LMO cathode presents conventional Li⁺ intercalation/de-intercalation behaviours in the HA electrolyte, Fig. 6c. The LMO cathode using the HA gel electrolyte maintains a comparable rate capability to that of ones using the liquid

electrolyte, making it suitable for high-power applications (Supplementary Fig. S43). More importantly, the hybrid AZMBs exhibit noticeable improvement in cycle stability, with a capacity retention of 82% after 1000 cycles (Supplementary Fig. S44), in contrast to those using the liquid electrolyte which fails after only 133 cycles due to short circuits (Supplementary Fig. S45). In addition, the significantly lower presence of ZHS by-products on the cycled Zn anode in the gel electrolyte further verifies the exceptional corrosion-inhibiting capability of the HA gel electrolyte (Supplementary Fig. S46). The short circuit is attributed to the growth of Zn dendrites and gas evolution. The gas evolution in the hybrid AZMBs was further evidenced using differential electrochemical mass spectrometry. Results from the cell with the liquid electrolyte reveal the presence of hydrogen towards the end of the charging process, with its amount increasing with cycling (Fig. 6d). In contrast, the HA gel electrolyte releases negligible hydrogen, indicating significant improvement in reduction stability provided by the HA gel electrolyte (Fig. 6e). Additionally, no OER is observed in both electrolytes, indicating good compatibility of the LMO cathode with $ZnSO_4$-based electrolytes (Supplementary Fig. S47). Furthermore, Zn//LMO cells were evaluated under more demanding conditions with a low N/P ratio (around 3.3) and a thin Zn metal (16 μm). In this case, Zn//LMO cells using the HA gel electrolytes exhibit excellent anti-corrosion performance and Zn deposition regulation capability, allowing the cells to operate for at least 300 cycles (Supplementary Fig. S48). In contrast, Zn//LMO cells operating in the liquid electrolyte only last for 45 cycles.

To further demonstrate the practical application potential of the HA gel electrolyte, we assembled single-layer Zn//LMO pouch cells, which exhibit a specific capacity of 93.9 mAh g$^{-1}$ (based on the cathode mass) at a current rate of 1 C with a capacity retention rate of 78% after 400 cycles, together with a high average Coulombic efficiency of 99.83%, suggesting its good electrochemical property (Fig. 6f, g). By virtue of the HA gel electrolyte, flexible Zn//LMO pouch cells were also fabricated (Supplementary Fig. S49), delivering a specific capacity of 111.0 mAh g$^{-1}$ at a current rate of 1 C. Furthermore, when subjected to different deformation states of bending (e.g., 90° and 180°), folding, and rolling, the flexible Zn//LMO cell delivered capacity retention of 101%, 98%, 94% and 90%, respectively, manifesting its good flexibility to provide stable energy to power the hydrometer (Fig. 6h, i and Supplementary Fig. S50). Additionally, we also assessed the potential of the HA gel electrolyte in Zn-iodine (I$_2$) batteries for wearable and implantable devices, considering the good safety and biocompatibility of iodine cathodes[72–74]. As shown in Supplementary Fig. S51, the I$_2$/I$^-$ redox reaction in Zn//I$_2$ batteries using the HA gel electrolytes is well maintained within a voltage window of 0.5–1.6 V, delivering a capacity of 194.3 mAh g$^{-1}$ at a current rate of 10 C and excellent capacity retention of 92% after 10,000 cycles and 73% after 20000 cycles (Supplementary Fig. S52a). In contrast, Zn//I$_2$ batteries using liquid electrolytes have a short lifespan of only around 2000 cycles, and the short circuit was induced by the Zn dendrite growth (Supplementary Fig. S52b). Because of the fast kinetics of the I$_2$ cathode, the cell using the HA gel electrolyte delivers an impressive specific capacity of 150.0 mAh g$^{-1}$ at an ultra-high current rate of 50 C (Supplementary Fig. S53), indicating its capability to provide a high power density. The high tailorability and impressive stretchability, combined with the excellent electrochemical performance of the HA gel electrolyte, make it a promising and reliable candidate for AZMBs in various fields, including sensors, wearable and implantable devices, and robotics.

## Discussion

In conclusion, we have presented the use of a biocompatible HA gel electrolyte as a highly promising approach for addressing the issues of Zn corrosion and regulating the Zn nucleation and growth behaviour. The HA glycosaminoglycan skeleton contains abundant hydrophilic functional groups, which reduces the reactivity of H$_2$O by forming strong hydrogen bonds with water. This mitigates the parasitic H$_2$ evolution reaction and $Zn_4SO_4(OH)_6 \cdot 3H_2O$ by-products formation. Moreover, the negatively charged HA molecule promotes the de-solvation of Zn$^{2+}$ by weakening the electrostatic force between Zn$^{2+}$ and H$_2$O/SO$_4^-$, leading to a long-lasting shielding effect and a uniform charge distribution on the Zn surface, thereby preventing Zn dendrite formation. The optimised Zn deposition behaviour in the HA gel electrolyte has resulted in the exceptional performance of assembled AZMBs, including high CE, high capacity, and stable cycling capabilities, which are among the best gel electrolyte-based AZMBs ever reported. Our study offers a promising gel chemistry that effectively regulates Zn behaviour, providing significant potential for biocompatible energy-related applications and beyond.

## Methods

### Electrolyte and electrode preparation

The $ZnSO_4 \cdot 7H_2O$ salt (purity ≥99.995%) and hyaluronic acid (HA) were purchased from Macklin and used directly without further treatment. The HA was chosen for its ubiquitous in the human body, which makes it crucial for many cellular and tissue functions. It has been in clinical use for over thirty years[75]. HA is a biocompatible material and can be rapidly turned over in the body by hyaluronidase, with tissue half-lives ranging from hours to days. Besides, the HA is in the form of a sodium salt and will dissociate into HA anions and Na$^+$ in the electrolyte. Though it was evidenced that the influence of free Na$^+$ on the Zn anode is not significant[76], the HA anions are expected to provide an electric shielding effect to regulate the deposition of the Zn anode. To prepare the 2 M $ZnSO_4$ liquid electrolyte, a specific amount of $ZnSO_4 \cdot 7H_2O$ was dissolved in deionized water and the solution was brought to volume using a volumetric flask. The HA gel electrolyte was prepared by mechanically mixing 6.0, 15.0 and 30.0 wt% HA powder with 2 M $ZnSO_4$ aqueous solvent at 40 °C overnight. The water content of as-prepared gel electrolytes is approximately 52.5, 47.5 and 39 wt%, respectively. The concentration of 6.0 wt% was an optimised value considering the balance of ionic conductivity and mechanical strength of the as-prepared gel electrolyte[77,78].

To prepare the LiMn$_2$O$_4$ (LMO) cathode electrode, LMO active material (Canrd Technology Co. Ltd.), super P and polytetrafluoroethylene (PTFE, dispersion in deionized water) were blended in a mass ratio of 6:2:2. The obtained electrode slurry was rolled in titanium (Ti) mesh current collectors (Carnd Technology Co. Ltd.) and then dried at 80 °C overnight to remove residue H$_2$O. For assembling flexible batteries, the LMO slurry was coated on carbon cloth to achieve good flexibility. For Zn//I$_2$ batteries, the catholyte was composed of 0.05 M I$_2$ and 1 M LiI in H$_2$O solution, where the host material was made of 75 wt% Ketjen black (KB) and 25 wt% PTFE on carbon cloths. The areal mass loadings of the LMO cathode and I$_2$ cathode were controlled at 11–12 and 2.0–2.5 mg cm$^{-2}$, respectively.

### Characterisations

The ionic conductivity (σ) of liquid electrolytes was measured using the 856 Conductivity Module (Sweden) at room temperature. To measure the conductivity of the HA gel electrolyte, a homemade symmetrical Ti//Ti cell was utilised, and the cell constant $K_{cell}$ was determined based on the ionic conductivity of the liquid electrolyte. The electrochemical impedance spectroscopy (EIS) of the Ti//Ti cell was conducted on a PGSTAT-30 machine (Autolab, Switzerland) with an amplitude of 5 mV and frequencies ranging from 500 kHz to 10 Hz. The ionic conductivity of the HA gel electrolyte is calculated according to the following equation (Eq. (3)):

$$\sigma = \frac{L}{SR} = \frac{K_{cell}}{R}$$

where $L$ is the distance between two Ti electrodes, $S$ is the contact area of two Ti electrodes, $R$ is the intersection of the experimental curve with the real impedance axis.

The mechanical properties of the hydrogel were evaluated using a universal testing system (Instron 5543). The tensile testing was conducted using rectangular specimens at a constant velocity of 200 mm min$^{-1}$. The compression testing was performed on cylindrical specimens at a constant velocity of 10 mm min$^{-1}$. Ex-situ Fourier transforms infra-red spectroscopic measurement was performed on a Nicolet 6700 ThermoFisher FTIR spectrometer in an attenuated total reflection (ATR) mode. Raman spectra were collected using a LabRAM HR Evolution Raman microscope (Horiba Jobin Yvon) with a 532 nm laser. The pH values of electrolytes were determined by a pH/Conductivity Multiparameter Benchtop Meter (Thermo Orion Versa Star Pro). X-ray diffraction (Rigaku Ultima IV) of Zn electrodes was conducted on monochromatic Cu Kα radiation, scanning between 5° and 80° at a rate of 10° min$^{-1}$. The morphological images and roughness of electrodes were acquired by scanning electron microscope (Hitachi SU7000) and conformal microscope (Olympus LEXT OLS5000 Profilometer).

### Electrochemical measurements

The electrochemical tests were conducted on CR 2032 coin-type cells using glass fibre filters as the separator. Zn foils with a thickness of 100 μm were used in cells if not mentioned, while the cells under 33% DoD$_{Zn}$ and 80% DoD$_{Zn}$ were conducted by using Zn foil with a thickness of ca. 16 μm. The total capacity of the used Zn foil ($C_{Zn}$) is calculated based on the mass of the Zn foil ($m_{Zn}$) according to the following equation (Eq. (4)):

$$C_{Zn} = m_{Zn} \times 820 \text{ mAh g}^{-1}$$

Before use, the Zn foil was polished by softback sanding sponges (3M, USA) and then wiped with ethanol. The cathodes and Zn anodes were cut as disk-shaped electrodes with a diameter of 12 mm for the coin cell assembly if not mentioned, while Zn anodes with a smaller diameter of 9 mm were used in the Zn//LMO full cells with a low N/P ratio of 3.3. The size of electrodes used in Zn//LMO pouch cells was 3 × 3 cm$^2$ and that in flexible Zn//LMO cells was 1 × 8 cm$^2$. Charge-discharge tests of coin and pouch cells were performed on LAND battery test system (CT2001A) and the Neware battery test system (CT-3008, China), respectively. For an intermittent charge/discharge test, Zn//Zn cells were successively charged/discharged at a current density of 1 mA cm$^{-2}$ with a total capacity of 1 mAh cm$^{-2}$ for 50 h and held at open circuit potential for 50 h. Zn//Cu asymmetric cells were charged-discharged at 1 mA cm$^{-2}$ with a capacity of 1 mAh cm$^{-2}$ and an upper cut-off voltage of 0.5 V to determine the Coulombic efficiency of different electrolytes. Zn//LMO full cells were cycled with a voltage range of 1.4–2.1 V (vs. Zn$^{2+}$/Zn) at a current rate of 1 or 3 C (1 C = 150 mAh g$^{-1}$). The voltage range of Zn//I$_2$ full cell was 0.5–1.6 V and the current rate was 10 C (1 C = 200 mAh g$^{-1}$). All battery tests were conducted at a constant temperature of 25 °C.

Electrochemical stability windows (ESW) of electrolytes were studied by linear sweep voltammetry tests on a platinum (Pt) electrode (∅ = 0.1 cm), with an Ag/AgCl reference electrode and a Pt counter electrode at a scan rate of 1 mV s$^{-1}$. Cyclic voltammetry of Zn//Cu cells was measured at a voltage range of −0.35 to 0.6 V (vs. Zn$^{2+}$/Zn) at 5 mV s$^{-1}$. Zn//Zn cells were measured at a voltage range of −15 to 15 mV at scanning rate from 4 to 12 mV s$^{-1}$. Tafel tests were conducted on the CHI760E electrochemical workstation using Zn//Zn cells at a scanning rate of 10 mV s$^{-1}$. The EIS of Zn//Zn cells was conducted with an amplitude of 5 mV and frequencies ranging from 100 kHz to 0.01 Hz. The real-time gas evolutions were observed by differential electrochemical mass spectrometry on the HPR-40 DEMS mass spectrometer system (HIDEN Analytical) with an ECC-Air cell (EL-CELL). The LMO//Zn

cell was tested at a voltage range of 1.4–2.1 V at 1 C. In situ synchrotron-based FTIR experiments were performed on the Infra-red Microspectroscopy (IRM) beamline at ANSTO – Australian Synchrotron (Clayton, Victoria), using a Hyperion 3000 FTIR microscope coupled to a Vertex 70/70 v FTIR spectrometer (Bruker Optik GmbH, Ettlingen, Germany), equipped with a customised Si crystal (250 μm). The test was performed on a Zn//stainless steel (SS) two-electrode setup using an ECC-Opto-Std-Aqu cell (EL-CELL) at a charge/discharge current density of 10 mA cm$^{-2}$ for 10 min. The tip of the Si crystal was embedded in the stainless steel mesh to collect the signal from the interface electrolyte layer[79]. The Zn$^{2+}$ transference number ($t_{Zn^{2+}}$) was measured by Bruce–Vincent method[80,81]. The Zn//Zn cells were subjected to constant potential (20 mV) for 120 min, and the impedance before and after the polarisation was recorded. The $t_{Zn^{2+}}$ was determined by the following equation (Eq. (5)):

$$t_{zn^{2+}} = \frac{I_{SS}(\Delta V - R_0 I_0)}{I_0(\Delta V - R_{SS} I_{SS})}$$

where $I_0$ and $I_{SS}$ are the initial and steady current, $R_0$ and $R_{SS}$ the initial and steady resistance, and $\Delta V$ the applied voltage (20 mV).

The de-solvation energy ($E_a$) was determined by measuring the impedance variation under different temperatures (0–50 °C) and then fitting the data according to the Arrhenius equation (Eq. (6)):

$$\frac{1}{R} = A \exp\left(-\frac{E_a}{RT}\right)$$

where $R$ is the interfacial impedance, $A$ is the pre-exponential factor, $R$ is the universal gas constant (8.314 J K$^{-1}$ mol$^{-1}$) and $T$ is the absolute temperature[68,82].

### Biocompatibility investigation

RAW 264.7 cells (American Type Culture Collection, ATCC) were cultured in Dulbecco's modified Eagle's medium (DMEM, ATCC) supplemented with 1% penicillin/streptomycin (PS) and 10% foetal bovine serum (FBS). HEKn cells (ATCC) were cultured in Keratinocyte SFM (Thermo Fisher) with human recombinant epidermal growth factor (rEGF, 0.15 ng mL$^{-1}$) and bovine pituitary extract (BPE, 25 μg mL$^{-1}$). Cell Proliferation Reagent water-soluble tetrazolium salt (WST-1, Roche, 11644807001) was used to assess the inhibitory effect of the HA gel electrolyte. Cells were cultured in a 96-well plate at 5 × 10$^3$ cells/well and incubated at 37 °C with 5% CO$_2$ and 95% air. Different amounts of HA gel electrolyte (0.1, 0.01 μg mL$^{-1}$) in DMEM or SFM were prepared and filtered with a 0.22 μm filter. The medium in 96-well plates was changed with HA gel electrolyte medium after being cultured overnight and then cultured for another 22 and 46 h. Cells treated with the pure medium were used as the control group. Subsequently, WST-1 was applied to detect the viability of HaCaT cells at 450 nm after 2 h using an Infinite 200 PRO plate reader system (Tecan). Experimental data were analysed using GraphPad Prism 9.0 (GraphPad Software). Comparisons between groups were analysed by one-way ANOVA followed by Dunnett's test as indicated. Experiment data are presented as mean ± standard deviation. A value of $P \leq 0.033$ was considered statistically significant.

### Computational methods

All MD simulations were performed using the GAFF2 force field[83]. The ACPYPE was employed to obtain the GAFF2 force field topology[84]. The simulation box size was 4 × 4 × 4 nm$^3$ for all simulation models, which consisted of Zn$^{2+}$, SO$_4^{2-}$, H$_2$O, with/without HA molecules. The cut-off distance of 1.2 nm was used for a Lennard–Jones potential. The Coulombic potential was measured using particle mesh Ewald (PME) with a cut-off distance of 1.2 nm and Fourier grid spacing of 0.12. All bonds were constrained with the LINCS algorithm. and periodic boundary

conditions were applied in all directions. The MD simulations were started by running initial energy minimisation, followed by 500 ps of NVT simulation and 500 ns of NPT simulation, with an integration time step of 0.001 ps. All simulation systems were finally maintained at 298 K using the Nose–Hoover thermostat for 70 ns to collect simulation data. A time constant of 1 ps was applied for the temperature coupling.

Several Zn-ion structures observed from the MD simulations were then taken for further investigation using density functional theory (DFT). The DFT calculations were implemented using the Vienna ab-initio simulation package (VASP)[85,86] with the core and valence electronic interactions being modelled using the projector augmented wave (PAW) method[87,88]. The Perdew–Burke–Ernzerhof (PBE) exchange-correlation function was employed[89]. The wavefunction was expanded with a kinetic energy cut-off of 500 eV and Gamma k-points were used. The dispersion correction was also considered in this study by using the DFT-D3 method[90].

## Data availability
The data generated in this study are provided in the Supplementary Information/Source data file. Source data are provided with this paper.

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

## Acknowledgements

This work was supported by the Australian Research Council (DP210101486 Z.G., FL210100050 Z.G.). S.Z. acknowledges the financial support from the Australian Research Council (DE240100159). Components of this research were undertaken on the IRM beamline at the Australian Synchrotron, part of ANSTO, through the merit-based beamtime proposals (M18819 S.Z., M19919 S.Z.). G.L. was supported by scholarships from the China Scholarship Council (Grant No. 202006750014). The authors would also like to thank the Australian Institute of Nuclear Science and Engineering (AINSE) for providing financial assistance through the Early Career Researcher Grant (ECRG S.Z.). J.A.Y. acknowledges the high-performance computing facilities provided by National Computational Infrastructure (NCI) Australia. The authors also thank Dr. Yue Hui for his suggestion for biochemistry tests.

## Author contributions

Conceptualisation: G.L., S.Z., Z.G. Experimental design and investigation: G.L., S.Z. Data analyses: G.L., Z.Z., L.S., M.L. J.H., J.M. Characterisation of synchrotron-based ATR: G.L., J.V. Theoretical simulation: J.A.Y. L.X. Biosecurity characterisation: Z.Z., C-X.Z. Writing—original draft: S.Z., G.L. Writing—review & editing: G.L., S.Z., J.H., Z.G.

## Competing interests

The authors declare no competing interests.
