## [Peer Review File · Nature Communications]

A Biocompatible Electrolyte Enables Highly Reversible Zn Anode for Zinc Ion BatteryREVIEWER COMMENTS

Reviewer #1 (Remarks to the Author):

The authors reported a biocompatible hydrogel electrolyte by adding HA in ZnSO₄ solutions. The interaction between HA and H₂O reduce the HER and by-products on Zn anode, and the HA can regulate the solvation structure of Zn²⁺ to prevent dendrite formation. The result is good. However, some critical issues should still be clarified, and so far, this paper can only be suitable for a more specific journal.

1. The ionic conductivities of 71.4 and 54.4 mS cm⁻¹ are both too high for typical 2M ZnSO₄ electrolyte.
2. How to prepare the Zn battery with HA electrolyte? The mechanical property of the HA electrolyte should be provided if it is used as gel electrolyte independently in Zn batteries. If assembled with GF, it provides no flexibility.
3. The authors attribute the formation of hydrogel to the strong hydrogen bonding between HA and H₂O in Figure 1a. Without ZnSO₄, can HA form hydrogel with water?
4. Figure 1c seems not the magnified view of Figure 1e. Otherwise, the difference is too small between HA and LE, indicating neglectable effect of HA on HER inhibition.
5. It seems no difference between Fig s3a and b.
6. The pH variation of the electrolyte during 12 d immersing can be provided to further confirm the reaction between electrolyte and Zn anode.
7. The authors used Rs in Fig S7, Re in S8 and Rb in S9. Which one is correct? This is not a nat. commun. level.
8. The calculation method for DOD should be provided. A 10 μm Zn usually cannot reach 6.5 mA h cm⁻².
9. If Zn₄SO₄(OH)₆·3H₂O (ZHS) is hexagonal plate, what is the loosely stacked Zn clusters?

Reviewer #2 (Remarks to the Author):

The author of this article presents a new, biocompatible, and cost-effective material, known as hyaluronic acid (HA), in conjunction with water and ZnSO₄ salt, to synthesize a hydrogel electrolyte with exceptional corrosion resistance towards zinc anodes, leading to notable battery performance. These unique gel properties offer tremendous potential for biocompatible energy-related applications and other fields. However, some key questions remain unanswered, and further revisions are recommended. In particular:

1. To supplement the current images in Figures S9 and S10, which demonstrate the performance and SEM images of the hydrogel electrolyte after the liquid solution has aged for 0, 7, and 12 days, the author may consider including cycling performance and SEM images of the HA hydrogel electrolyte.
2. It would be beneficial for readers' understanding if the author provided additional optical photos, including corresponding surface state diagrams of the HA hydrogel electrolyte on the zinc electrode or assembly diagrams of the HA hydrogel electrolyte in a button battery.
3. To further clarify the research, a simulation model diagram in Figure S9 of the Supporting Information could demonstrate the combination of the dominating cluster (Zn(H₂O)₆²⁺) with other clusters, such as ZnSO₄(H₂O)₅ and ZnSO₄(H₂O)₄, in the solvent structure.

4. The inclusion of optical photos of the HA hydrogel at different stretching degrees would provide a more detailed understanding of its properties.

5. I suggest that the graphs and corresponding units be checked more carefully; for example, paragraph 3 of the main text includes a superscript error whereby "mA cm⁻³" should read as "mA h cm⁻²."

Reviewer #3 (Remarks to the Author):

The development of batteries that possess both biocompatibility and long calendar life is crucial for the advancement of implantable devices in the future. This study by Li et al. reports a novel biocompatible hydrogel electrolyte based on hyaluronic acid (HA) to address the major challenges associated with aqueous zinc-based batteries for flexible and implantable devices. Those challenges include limited cycle life, side reactions, and biocompatibility. Compared to conventional liquid and other gel electrolytes, this HA gel electrolytes offer enhanced biosecurity with improved electrochemical performance, especially under a high DoD value of 80%, making it a promising alternative. The authors also conducted comprehensive investigations into the underlying mechanism of the HA gel electrolyte, including operando, spectral, and theoretical analyses. Therefore, this work serves as a valuable example and provides important theoretical insights for the progress of this field. In this case, it is recommended for publication in Nature Communications.

However, some clarifications are necessary to strengthen the findings and conclusions.

1. The measurement of the ESW of electrolytes is commonly performed on Ti or GC electrodes due to their high HER overpotential. In this study, the authors employed Pt electrode as the working electrode, different from this common practice. The rationale behind this choice should be clarified.
2. The HA electrolyte exhibits a mitigating effect on the corrosion of the Zn anode. To provide a comprehensive evaluation of this improvement, it is recommended to include the actual corrosion rate of Zn anodes in both electrolytes, instead of solely relying on the corrosion current as the sole metric.
3. The authors claimed that the coordination of Zn²⁺ and SO₄²⁻ is weaker in the HA gel electrolyte, based on findings from FTIR and DFT calculation. However, to strengthen this claim, it is recommended that the authors provide further evidence to substantiate it.
4. The authors should explain the significance of the increased Zn²⁺ transference number in the Zn plating/stripping process in the presence of the HA gel electrolyte.
5. The performance of Zn//LMO cells were evaluated under a demanding condition with a low N/P ratio (around 3.3). What is the cycle performance of the Zn//Zn symmetric cell under this condition?
6. In addition to capacity and cycle life, the rate performance of full cells is also of great significance. To provide a comprehensive evaluation, it is recommended to include the rate performance data for the full cells.
7. Some scratches can be observed on the Zn surface in Figure 2e. Did the authors conduct the pre-treatment on the Zn electrodes? If yes, it is important to provide experimental details regarding the preparation of the Zn anode.
8. A thorough revision of the entire manuscript is recommended to address typographical errors and inconsistencies. For example, the ZnSO₄(HA)(H₂O)₄ should be ZnSO₄(HA)(H₂O)₄– in the legend of Figure S15.

RESPONSE TO REVIEWS

Reviewer #1 (Remarks to the Author):

The authors reported a biocompatible hydrogel electrolyte by adding HA in ZnSO₄ solutions. The interaction between HA and H₂O reduces the HER and by-products on Zn anode, and the HA can regulate the solvation structure of Zn²⁺ to prevent dendrite formation. The result is good. However, some critical issues should still be clarified, and so far, this paper can only be suitable for a more specific journal.

Response:

We thank the Reviewer #1 for the positive comment on our work that ‘the result is good’. We also acknowledge that further clarifications on some critical aspects is necessary to improve the quality of our manuscript.

In response to all raised comments, we have carefully revised the manuscript by providing additional experimental data and analyses to strengthen our findings. With these revisions and clarifications, we believe that the manuscript could contribute to the field of aqueous zinc metal batteries (AZMBs) and hydrogel electrolytes. The followings are key features of our work:

- 1) The point of our work is the development of biocompatible hyaluronic acid (HA)-based hydrogel electrolyte for batteries used in soft wearable and implantable electronic devices. This hydrogel addresses critical challenges faced by current organic-electrolyte-based lithium ion batteries (*e.g.*, potential explosion and hazards related to electrolyte leakage) and hydrogel-electrolyte-based AZMBs (*e.g.*, inferior electrochemical performance and short calendar life).
- 2) We conducted performance evaluations of the proposed HA gel electrolyte by substantial electrochemical testing, particularly with a high Zn utilisation rate of 80%, in order to mimic the practical testing conditions. Our findings demonstrate excellent performance of the HA electrolyte, both in half-cell and full-cell configurations.
- 3) We performed electrochemical analyses, *in situ* spectroscopic techniques and theoretical computations to understand the underlying mechanism of the HA molecule on the enhanced performance of the AZMBs.

Our findings present a hydrogel with both safe and high performance for wearable and implantable devices and provide new insights in the design of practical hydrogel electrolytes. We think that this work will be of great interest to both researchers and industry, appealing to the readership of *Nature Communications*.

Comment 1-1

The ionic conductivities of 71.4 and 54.4 mS cm⁻¹ are both too high for typical 2M ZnSO₄ electrolyte.

Response:

We agree with the Reviewer #1 that the observed ionic conductivities of 71.4 mS cm⁻¹ (2M ZnSO₄) and 54.4 mS cm⁻¹ (the HA gel electrolyte) in our manuscript are higher than those of typical ZnSO₄-based electrolytes. The reason is that those ionic conductivities were accidentally measured in a laboratory, where the ambient temperature surpassed 25 °C. As a result, this environment contributes to the increase in the observed ionic conductivity levels.

To mitigate the potential influence of temperature, we conducted additional experiments under a controlled standard ambient condition (at 25 °C with a humidity level of 30±5%), which is similar to our

electrochemical testing condition illustrated in our original manuscript. This standard condition was applied to determine the ionic conductivity of both the 2M ZnSO₄ electrolyte and the HA gel electrolyte. To enhance the accuracy and reliability of the data, we employed two kinds of testing methods, namely, using *the conductivity meter* and *the impedance* methods, to ensure a rigid assessment of the ionic conductivity. As is shown in **Figure R1**, ionic conductivities of the 2M ZnSO₄ liquid electrolyte and HA gel electrolyte are measured to be 57.0 and 47.7 mS cm⁻¹, respectively, which align well with previous reports for the 2M ZnSO₄ aqueous electrolyte (range from 55 to 58 mS cm⁻¹).^[1] Additionally, the HA gel electrolyte exhibits a superior conductivity value when compared with those of other gel electrolytes, as shown in **Table R1**.

Figure R1. Nyquist plots of Ti//Ti symmetrical cells: (a) using the 2M ZnSO₄ liquid electrolyte and (b) the HA gel electrolyte. (c) Diagram of comparison of ionic conductivity between the 2M ZnSO₄ liquid electrolyte and the HA gel electrolyte. The error bars represent the standard deviation of the values.

Table R1. The comparison of ionic conductivity with previously reported gel electrolytes. Abbreviations of gel electrolytes are as follows: Hyaluronic acid (HA), poly(ethylene glycol) (PEG), poly(acrylamide-co-[2-(methacryloyloxy)ethyl]dimethyl-(3-sulfopropyl)) (PASHE), carboxyl-grafted polyvinyl alcohol and xanthan gum (PSX), poly(3-(1-vinyl-3-imidazolium) propanesulfonate) (PVIPS), poly-2-Acrylamido-2-methylpropanesulfonic/polyacrylamide (PAMPS-PAAM), polyacrylamide-poly (ethylene glycol) diacrylate-carboxymethyl cellulose (PMC), poly 2-acrylamido-2-methyl-1-propane sulfonate zinc (PAMPSZn), iota-carrageenan (IC), polyacrylamide (ZS/GL/AN), sorbitol-modified cellulose (Sor-Cel).

Gel electrolytes	Salt	Solvent	Ionic conductivity (mS cm ⁻¹)	Ref.
HA	2M ZnSO ₄	H ₂ O	47.7	This work
PEG	Zn(ClO ₄) ₂	/	12.6	[2]
PASHE	1M ZnSO ₄	H ₂ O	32.9	[3]
PSX	2M ZnSO ₄	H ₂ O	18.9	[4]
PVIPS	2M ZnSO ₄ + 0.1M MnSO ₄	H ₂ O	21.9	[5]
PAMPS-PAAM	ZnSO ₄	DMSO	21.6	[6]
PMC	2M ZnSO ₄	H ₂ O	30.2	[7]
PAMPSZn	Zn(OH) ₂ •2ZnCO ₃	H ₂ O	15.6	[8]
IC	2M ZnSO ₄	H ₂ O	43.0	[9]
ZS/GL/AN	3M ZnSO ₄	H ₂ O	13.9	[10]
Sor-Cel	16M ZnCl ₂ + 0.6M CaCl ₂	H ₂ O	35.4	[11]

In response to address fully this comment, we have in our R-MS and R-SI, included following revision, namely:

1) R-SI, p. 10-11, included Figure S3, Table S1, namely:

Figure S3. Nyquist plots of Ti//Ti symmetrical cells: (a) using the 2M ZnSO₄ liquid electrolyte and (b) the HA gel electrolyte. (c) Diagram of comparisons of ionic conductivity between the 2M ZnSO₄ liquid electrolyte and the HA gel electrolyte. The error bars represent the standard deviation of the values.

Table S1. The comparison of ionic conductivity with previously reported gel electrolytes. Abbreviations of gel electrolytes are as follows: Hyaluronic acid (HA), poly(ethylene glycol) (PEG), poly(acrylamide-co-[2-(methacryloyloxy)ethyl]dimethyl-(3-sulfopropyl)) (PASHE), carboxyl-grafted polyvinyl alcohol and xanthan gum (PSX), poly(3-(1-vinyl-3-imidazolium) propanesulfonate) (PVIPS), poly-2-Acrylamido-2-methylpropanesulfonic/polyacrylamide (PAMPS-PAAM), polyacrylamide-poly (ethylene glycol) diacrylate-carboxymethyl cellulose (PMC), poly 2-acrylamido-2-methyl-1-propane sulfonate zinc (PAMPSZn), iota-carrageenan (IC), polyacrylamide (ZS/GL/AN), sorbitol-modified cellulose (Sor-Cel).

Gel electrolytes	Salt	Solvent	Ionic conductivity (mS cm⁻¹)	Ref.
HA	2M ZnSO₄	H₂O	47.7	This work
PEG	Zn(ClO₄)₂	/	12.6	[13]
PASHE	1M ZnSO₄	H₂O	32.9	[14]
PSX	2M ZnSO₄	H₂O	18.9	[15]
PVIPS	2M ZnSO₄ + 0.1M MnSO₄	H₂O	21.9	[16]
PAMPS-PAAM	ZnSO₄	DMSO	21.6	[17]
PMC	2M ZnSO₄	H₂O	30.2	[18]
PAMPSZn	Zn(OH)₂•2ZnCO₃	H₂O	15.6	[19]
IC	2M ZnSO₄	H₂O	43.0	[20]
ZS/GL/AN	3M ZnSO₄	H₂O	13.9	[21]
Sor-Cel	16M ZnCl₂ + 0.6M CaCl₂	H₂O	35.4	[22]

2) R-MS, p. 7, included in-text discussion, namely:

‘The measured ionic conductivity value for the HA gel electrolyte is slightly lower than that of the liquid electrolyte, decreasing from 57.0 mS cm⁻¹ to 47.7 mS cm⁻¹ (Figure S3), but it still outperforms other reported gel electrolytes (Table S1) and is sufficient to provide rapid ion migration.’

Comment 1-2

How to prepare the Zn battery with HA electrolyte? The mechanical property of the HA electrolyte should be provided if it is used as gel electrolyte independently in Zn batteries. If assembled with GF, it provides no flexibility.

Response:

We confirm that the HA gel electrolyte was directly used in our assembled Zn batteries without using separators. The schematic illustration can be found in **Figure R2**.

To assess mechanical properties of the HA gel electrolyte, we conducted additional experiments, including tests to determine the tensile strength and compressive strength. These results are shown in **Figure R3a** and **b**, respectively. The HA gel electrolyte exhibits a moderate elongation-at-break of 220%

and a compressive strength of 0.18 MPa. Also, the HA gel electrolyte can be stretched from 2 cm to 6 cm, representing a 200% increase from the original length, without any mechanical failure observed (**Figure R3c**). Our findings indicate that the HA gel electrolyte possesses good mechanical properties, which is suitable for flexible applications. These values are also comparable or even exceed some of previously reported gel electrolytes, as summarised in **Table R2**.

Additionally, it was observed that, by reducing the water content in the HA gel electrolyte, the mechanical strength of the HA hydrogel can be further enhanced, which is promising for other applications that require high mechanical strength and stability (**Figure R4**). Noted, this enhancement may be accompanied by a minor trade-off between electrochemical performance and mechanical properties, due to a slight reduction in ionic conductivity.

Figure R2. Schematic diagram of flexible Zn/LMO battery using the HA gel electrolyte. An optical photo (below) shows the cathode/hydrogel electrolyte/Zn configuration.

Figure R3. (a) Tensile and (b) compressive stress–strain curves of the HA gel electrolyte. (c) Optical photos showing the stretching process of the HA gel electrolyte.

Figure R4. (a) Tensile and (b) compressive stress–strain curves and (c) ionic conductivity of HA gel electrolytes with different water contents.

Table R2. The comparison of tensile strength and compressive strength with previously reported gel electrolytes. Abbreviations of gel electrolytes are as follows: Hyaluronic acid (HA), poly(acrylamide-co-[2-(methacryloyloxy)ethyl]dimethyl-(3-sulfopropyl)) (PASHE), poly(3-(1-vinyl-3-imidazolium)propanesulfonate) (PVIPS), polyacrylamide-poly (ethylene glycol) diacrylate-carboxymethyl cellulose (PMC), iota-carrageenan (IC), polyacrylamide (ZS/GL/AN), sorbitol-modified cellulose (Sor-Cel).

Gel electrolytes	Elongation-at-break (%)	Compressive strength (MPa)	Compressive strain (%)	Ref.
HA-6%	220	0.18	80%	This work
HA-15%	251	0.39	80%	This work
HA-30%	333	0.81	80%	This work
PASHE	315	0.05	60%	[3]
PVIPS	62	/	/	[5]
PMC	100	/	/	[7]
IC	128	/	/	[9]
ZS/GL/AN	350	/	/	[10]
Sor-Cel	303	0.085	50%	[11]

In response to address fully this comment, we have in our R-MS and R-SI, included following revision, namely:

- 1) R-SI, *p.* 12-14, included Figure S4, Figure S5 and Table S2, namely:

Figure S4. (a) Tensile and (b) compressive stress–strain curves of the HA gel electrolyte. (c) Optical photos showing the stretching process of the HA gel electrolyte.

Figure S5. (a) Tensile and (b) compressive stress–strain curves and (c) ionic conductivity of HA gel electrolytes with different water contents.

Table S2. The comparison of tensile strength and compressive strength with previously reported gel electrolytes. Abbreviations of gel electrolytes are as follows: Hyaluronic acid (HA), poly(acrylamide-co-[2-(methacryloyloxy)ethyl]dimethyl-(3-sulfopropyl)) (PASHE), poly(3-(1-vinyl-3-imidazolio)propanesulfonate) (PVIPS), polyacrylamide-poly (ethylene glycol) diacrylate-carboxymethyl cellulose (PMC), iota-carrageenan (IC), polyacrylamide (ZS/GL/AN), sorbitol-modified cellulose (Sor-Cel).

Gel electrolytes	Elongation-at-break (%)	Compressive strength (MPa)	Compressive strain (%)	Ref.
HA-6%	220	0.18	80%	This work
HA-15%	251	0.39	80%	This work
HA-30%	333	0.81	80%	This work
PASHE	315	0.05	60%	[14]
PVIPS	62	/	/	[16]
PMC	100	/	/	[18]
IC	128	/	/	[20]
ZS/GL/AN	350	/	/	[21]
Sor-Cel	303	0.085	50%	[22]

2) R-MS, p. 7, included in-text discussion, namely:

‘Figure S4a and b illustrates that the HA gel material displays a moderate elongation-at-break of 220% and a compressive strength of 0.18 MPa. The HA gel electrolyte can be stretched from 2 cm to 6 cm, representing a 200% increase in its original length, without any mechanical failure (Figure S4c). Additionally, by reducing the water content in the hydrogel, the mechanical strength of the HA gel electrolyte can be further enhanced for applications demanding high mechanical stability, at a minor cost of slightly decreased ionic conductivity (Figure S5). As summarised in Table S2, the mechanical strength of the HA gel electrolyte is comparable or even exceed some of previously reported gel electrolytes, indicating its suitability as a candidate for flexible applications.’

3) R-SI, p. 61, included Figure S48.

Figure S48. Schematic diagram of flexible Zn/LMO battery using the HA gel electrolyte. An optical photo (below) shows the cathode/hydrogel electrolyte/Zn configuration.

Comment 1-3

The authors attribute the formation of hydrogel to the strong hydrogen bonding between HA and H₂O in Figure 1a. Without ZnSO₄, can HA form hydrogel with water?

Response:

Yes, the HA can form hydrogel with bare water and without the need for ZnSO₄ salt addition. As shown in **Figure R5**, the HA can easily form a hydrogel when mixed with bare water (HA: water ratio is 1:9, in weight), in the absence of ZnSO₄ salt. This result indicates that the hydrogel formation in our HA gel electrolyte is irrelevant to the ZnSO₄ salt. Additionally, the HA can be hydrogel forms with other aqueous electrolytes containing different salts, such as zinc chloride (ZnCl₂) and zinc trifluoromethyl sulfonate (Zn(OTF)₂) (**Figure R5b-c**), which demonstrates the versatility of the HA molecule in hydrogel formation.

It is also noted that the presence of strong hydrogen bonds in HA-water hydrogel exhibits a noticeable increase, rising from 58.5% (in pure water) to 63.1% (HA-water, **Figure R6**). This finding suggests that strong hydrogen bonding interactions is formed between HA and water molecules, resulting in the formation of the hydrogel.

Figure R5. Optical photos of (a) the HA-H₂O hydrogel, (b) the HA-ZnCl₂ gel electrolyte and (c) the HA-Zn(OTF)₂ gel electrolyte.

Figure R6. FTIR spectra and related curve fitting results in the range of 2400–4000 cm^{-1} observed in (a) pure water and (b) HA-H₂O gel electrolyte. (c) Ratios of different hydrogen bonds calculated from fitting results of FTIR spectra in (a) and (b).

In response to address fully this comment, we have in our R-MS and R-SI, included following revision, namely:

1) R-SI, p. 9, included Figure S2, namely:

Figure S2. Optical photos of (a) the HA-H₂O hydrogel, (b) the HA-ZnCl₂ gel electrolyte and (c) the HA-Zn(OTF)₂ gel electrolyte.

2) R-MS, p. 6, included in-text discussion, namely:

‘The gelation of HA-ZnSO₄-H₂O system is primarily attributed to the strong hydrogen bonding between HA and water, as evidenced by the formation of HA-H₂O gel in the absence of zinc salt (Figure S2). The gel formation is not limited to ZnSO₄ and can also occur in other zinc salt systems, such as zinc chloride and zinc trifluoromethyl sulfonate, which demonstrates a broader applicability of the HA in different zinc salt systems.’

3) R-SI, p. 17, included Figure S8, namely:

Figure S8. FTIR spectra and their curve fitting results in the range of 2400–4000 cm^{-1} observed in (a) pure water and (b) HA-H₂O gel electrolyte. (c) Ratios of different hydrogen bonds calculated from fitting results of FTIR spectra in (a) and (b).

4) R-MS, p. 7, included in-text discussion, namely:

‘The fitting results in **Figure S7c** demonstrate that the HA gel electrolyte has a higher percentage of strong hydrogen bonds (87.2%) compared to that of the liquid electrolyte (81.5%). Additionally, it is noted that the presence of strong hydrogen bonds in HA-water hydrogel also exhibits a noticeable increase, rising from 58.5% (in pure water) to 63.1% (HA-H₂O, **Figure S8**). These findings indicate that HA has a strong coordination capability with H₂O, which is further supported by Raman spectra shown in **Figure S9** and **S10**.’

Comment 1-4

Figure 1c seems not the magnified view of *Figure 1e*. Otherwise, the difference is too small between HA and LE, indicating neglectable effect of HA on HER inhibition.

Response:

We apologise for the confusion caused by improper scaling of the data plots. The overlage current for Zn deposition masked the current for HER in original **Figure 1e**. However, we would like to clarify that **Figure 1c** in the original manuscript is the magnified view of the data presented in **Figure 1e**. To convey the accuracy and reliability of our results, we have provided the original data file obtained from the *Biologic* electrochemical workstation and the plotted data file by using *Origin* software, which are named as *Raw data of HER test* and *Plotted data of HER test*, respectively. In the *Origin* data, the unit of current has been converted to current density based on the area of the Pt electrode ($\phi=0.5$ mm).

To accurately present our data, we have adjusted the Y-scale, as shown in **Figure R7**. The Y-axis unit has also been converted from “A cm^{-2} ” to “mA cm^{-2} ”. **Figure R7a** clearly demonstrates that the HER potential was extended by 0.16 V (from -0.36 V to -0.52 V) in the presence of the HA gel electrolyte, resulting in a reduction in the responsive current density at -1.0 V. These observations are consistent with the data summarised in **Table R3**, which highlights the improvements attained by the HA gel electrolyte compared to previously reported gel electrolytes.

Furthermore, as is shown in **Figure R8**, our Differential Electrochemical Mass Spectrometry (DEMS) analysis reveals a significant suppression in the generation of H₂ gas during the charge/discharge cycles of Zn/LMO full cells when employing the HA gel electrolyte. This observation provides further evidence of the effective inhibition of the HER by the HA gel electrolyte.

Based on these findings, we confidently believe that the HA gel electrolyte exhibits a capability to suppress the HER process, making it as a promising candidate for advanced ZIBs applications.

Figure R7. Magnified views of the regions outlined near (a) anodic and (b) cathodic extremes in (c) overall electrochemical stability window of liquid and HA gel electrolytes on non-active Pt electrodes.

Table R3. The comparison of improvement of HER potential with previously reported gel electrolytes. Abbreviations of gel electrolytes are as follows: Hyaluronic acid (HA), poly(ethylene glycol) (PEG), poly(acrylamide-co-[2-(methacryloyloxy)ethyl]dimethyl-(3-sulfopropyl)) (PASHE), iota-carrageenan (IC), polyacrylamide (ZS/GL/AN).

Gel electrolytes	Expansion of HER potential (V)	Working electrode	Ref.
HA	0.16	Pt	This work
PEG	0.04	Zn	[2]
PASHE	0.02	Pt	[3]
IC	0.20	/	[9]
ZS/GL/AN	0.11	Stainless steel	[10]

Figure R8. Differential Electrochemical Mass Spectrometry spectra of Zn/LMO pouch cells upon charge/discharge at a current density of 1C in: (a) the liquid electrolyte and (b) the HA gel electrolyte.

In response to address fully this comment, we have, respectively, in our R-MS, *p.* 6 and *p.* 8, included Figure 1 and in-text explanation, namely:

Figure 1. Characterisation of physical properties and electrochemical stable window of the HA gel electrolyte. (a) Optical photos of liquid and HA gel electrolytes. Liquid electrolyte: 2M ZnSO₄ aqueous solution. (b) Calculated number of H-bond of single H₂O and HA molecule and binding energies of H₂O–H₂O and H₂O–HA. Magnified views of the regions outlined near (c) anodic and (d) cathodic extremes in (e) overall electrochemical stability window of liquid and HA gel electrolytes on non-active Pt electrodes.

*'The potentials corresponding to a current density of 0.1 mA cm^{-2} are defined as the onset potentials for hydrogen evolution reaction (HER) and oxygen evolution reaction (OER). Due to the highly interconnected hydrogen bonds network in the HA gel electrolyte, the reactivity of H_2O is effectively suppressed, resulting in a lower potential for HER (-0.52 V vs. -0.36 V , **Figure 1c**). Concurrently, the potential of the OER shows a slight increase, accompanied by a reduced current response (**Figure 1d**).'*

Comment 1-5

It seems no difference between Fig s3a and b.

Response:

To better illustrate the difference between **Figure S3a** and **b** in original manuscript (has been changed to **Figure S7 in R-SI**), we have plotted them together in one single figure (**Figure R9**). The comparison clearly reveals a noticeable shift towards lower wavenumbers in the FTIR spectrum of the HA gel electrolyte, indicating a difference in chemical bonding.

Additionally, by using molecular dynamics (MD) simulations, we investigated the binding behaviour of H_2O molecules within the liquid electrolyte and the HA electrolyte. **Figure R10** indicates that a 6.9% (from 7200 to 6700) decrease in the number of hydrogen bonds between H_2O molecules within the HA gel electrolyte, when compared with that of liquid electrolyte. This can be explained by the increase of the strong hydrogen bonds between HA and H_2O molecular in the HA gel electrolyte observed in FTIR spectra (**Figure R11**, the **Figure S7 in R-SI**). These complementary findings provide solid evidence, confirming that a partial of weak H_2O - H_2O bonds is transformed into strong HA- H_2O bonds within the HA gel electrolyte.

Figure R9. FTIR spectra of liquid and HA gel electrolytes in the selected range.

Figure R10. The amount of hydrogen bonds between H₂O and H₂O in liquid and HA gel electrolytes by using theoretical calculations.

Figure R11. FTIR spectra and related curve fitting results in the range of 2400–4000 cm⁻¹ observed in the (a) liquid electrolyte and (b) HA gel electrolyte. (c) Ratios of different hydrogen bonds calculated from fitting results of FTIR spectra in (a) and (b).

In response to address fully this comment, we have in our R-MS and R-SI, included following revision, namely:

- 1) R-SI, p. 20, included Figure S11, namely:

Figure S11. The amount of hydrogen bonds between H₂O and H₂O in liquid and HA gel electrolytes by using theoretical calculations.

2) R-MS, p. 8, included in-text discussion, namely:

*'The presence of HA in the gel electrolyte leads to a competition between HA and H₂O molecules for hydrogen bonding, resulting in a reduction of hydrogen bonds between H₂O and H₂O by 6.9% (from 7200 to 6700, **Figure S11**). This decrease is consistent with the increase of strong hydrogen bonds by 5.7% (HA-H₂O binding) in the HA gel electrolyte observed by FTIR spectra (**Figure S7**), indicating a transformation of H₂O-H₂O bonds into HA-H₂O bonds.'*

Comment 1-6

The pH variation of the electrolyte during 12 d immersing can be provided to further confirm the reaction between electrolyte and Zn anode.

Response:

We agree with Reviewer #1 that the investigation of pH variation in the electrolyte during immersion is needed, which help to confirm the corrosion reaction between the electrolyte and Zn anode.

As is shown in **Figure R12**, when Zn metal was immersed in the liquid electrolyte, the pH value of the liquid electrolyte rapidly increased from 4.25 to 5.20 within 3 days, indicating the drastic corrosive reaction in the liquid electrolyte. The pH value was continuously increased to 5.45 after soaking for 12 days in the liquid electrolyte, resulting in the formation of a significant amount of Zn₄SO₄(OH)₆·3H₂O (ZHS) by-product with a hexagonal plate structure, which deposited on the surface of the Zn electrode. In contrast, negligible variation in pH value was observed when Zn was immersed in the HA gel electrolyte, demonstrating an exceptional corrosion tolerance capability of the hydrogel.

Figure R12. The pH variation of liquid and HA gel electrolytes with Zn anodes immersed in each electrolyte for different days.

In response to address fully this comment, we have in our R-MS and R-SI, included following revision, namely:

- 1) R-SI, p. 24, included Figure S14, namely:

Figure S14. The pH variation of liquid and HA gel electrolytes with Zn anodes immersed in each electrolyte for different days.

2) R-MS, p. 10, included in-text discussion, namely:

'Upon immersion of Zn metal in the liquid electrolyte, the pH value of the electrolyte rapidly increases from 4.25 to 5.20 within 3 days, indicating the corrosive nature of the electrolyte (Figure S14). The pH value was continuously increased to 5.45 after soaking 12 days, resulting in the formation of a significant amount of $Zn_4SO_4(OH)_6 \cdot 3H_2O$ (ZHS) by-product with a hexagonal plate structure, which deposited on the surface of the Zn electrode (Figure 2c, d and S15). The formation of ZHS by-product results from the complexation reaction between Zn^{2+}/SO_4^{2-} and OH^- that remains in the electrolyte during the corrosion of the Zn anode by H^+ and subsequent release of H_2 from the electrolyte.^[18] In contrast, negligible pH variation and less by-products were observed when Zn was immersed in the HA gel electrolyte, indicating an exceptional corrosion tolerance capability (Figure 2e and S15).'

Comment 1-7

The authors used R_s in Fig S7, R_e in S8 and R_b in S9. Which one is correct? This is not a nat. commun. level.

Response:

We apologise for the careless in using the incorrect acronym for electrolyte resistance throughout our manuscript. We have thoroughly reviewed and proofread the manuscript to ensure the correct acronym and formatting are consistently used throughout.

In response to address fully this comment, we have, respectively, in our R-SI, p. 26 and p. 28, included revised Figure S16 and Table S4, namely:

Figure S16. Nyquist plots of Zn//Zn symmetric cells when immersing in liquid and HA gel electrolytes for (a) 0h, (b) 6h, (c) 24h, (d) 3 days and (e) 7days, and (f) corresponding fitting equivalent circuit.

Table S4. Fitting results of interfacial impedance of Zn//Zn symmetric cells when immersing in liquid and HA gel electrolytes.

Time	R_e (Ω)		R_f (Ω)		R_{ct} (Ω)	
	Liquid electrolyte	HA gel electrolyte	Liquid electrolyte	HA gel electrolyte	Liquid electrolyte	HA gel electrolyte
0h	3.8	3.7	38.6	82.3	170.1	153.8
6h	4.8	5.0	280.0	60.7	811.3	470.9
24h	6.5	3.7	387.3	272.0	1789.0	633.6
3d	15.0	4.9	989.6	508.2	4003.0	901.5
7d	15.7	3.2	1855.0	402.7	7740.0	1463.0

Comment 1-8

The calculation method for DOD should be provided. A 10 μm Zn usually cannot reach 6.5 mA h cm⁻².

Response:

We acknowledge Reviewer #1's comment regarding the capacity of a 10 μm Zn foil and appreciate his/her attention to the experimental details.

To address the concern, we conducted further measurements of the Zn foil thickness used in our experiments. **Figure R13a** and **b** demonstrate that the actual thickness of the Zn foil is approximately 16 μm, despite being labelled as 10 μm by the supplier. This has been correctly labelled in our R-MS.

Fortunately, we utilised the mass of the Zn foil (m_{Zn}) to calculate the total capacity (C_{Zn}) in our original manuscript by using the following equation:

$$C_{Zn} = m_{Zn} \times 820 \text{ mA h g}^{-1}$$

Figure R13c presents the weight of ten polished Zn foils with an identical diameter of 12 mm, from which we calculated the mass of one Zn foil to be 11.21 mg. By using this value, the areal capacity of Zn can be determined to be 8.13 mA h cm⁻². Therefore, considering an 80% depth of discharge (DoD) condition, the corresponding areal capacity would be 6.5 mA h cm⁻², which was used in our Zn//Zn symmetric cell test (**Figure 5h**).

Figure R13. The thickness of Zn foil obtained from (a) a micrometre and (b) cross-sectional SEM image. (c) The total mass of ten Zn foils.

In response to address fully this comment, we have in our R-SI, p. 4, included following revision, namely:

‘Zn foils with a thickness of 100 μm were used in cells if not mentioned, while the cells under 33% DoD_{Zn} and 80% DoD_{Zn} were conducted by using Zn foil with a thickness of ca. 16 μm. The total capacity of the used Zn foil (C_{Zn}) is calculated based on the mass of the Zn foil (m_{Zn}) according to the following equation:

$$C_{Zn} = m_{Zn} \times 820 \text{ mA h g}^{-1}$$

Before use, the Zn foil was polished by softback sanding sponge (3 M, U.S.A.) and then wiped with ethanol.’

Comment 1-9

If Zn₄SO₄(OH)₆·3H₂O (ZHS) is hexagonal plate, what is the loosely stacked Zn clusters?

Response:

The hexagonal plates observed in SEM image (**Figure R14**) are identified as the by-product, namely, Zn₄SO₄(OH)₆·3H₂O (ZHS), which exhibits a large size range of 20 – 100 μm. In contrast, the loosely stacked Zn clusters observed on Cu current collectors (**Figure R15**) have a smaller size range of 0.5 – 5 μm. Additionally, only a few amount of ZHS was observed on the Zn anode after aging for 3 days (**Figure R14b**). The deposition time (Zn plating on Cu) is no more than 10 hours, which is insufficient for the formation of considerable amount of ZHS by-product derived from corrosion reactions in the liquid electrolyte. Based on these observations, if the reviewer #1 commented on the loosely stacked Zn clusters in **Figure R15**, we conclude that the loosely stacked Zn clusters should be newly deposited Zn particles, rather than the ZHS by-product.

Figure R14. SEM image of ZHS by-products on the Zn anode before and after immersion in the liquid electrolyte for different days.

Figure R15. SEM images of plated Zn on Cu current collectors at a constant current density of 1 mA cm^{-2} with different areal capacities in the liquid electrolyte.

In response to this comment of Reviewer #1, no change has been made to our R-MS.

Reviewer #2 (Remarks to the Author):

The author of this article presents a new, biocompatible, and cost-effective material, known as hyaluronic acid (HA), in conjunction with water and ZnSO₄ salt, to synthesize a hydrogel electrolyte with exceptional corrosion resistance towards zinc anodes, leading to notable battery performance. These unique gel properties offer tremendous potential for biocompatible energy-related applications and other fields. However, some key questions remain unanswered, and further revisions are recommended. In particular:

Response:

We thank Reviewer #2 for his /her positive feedback and constructive comments.

Comment 2-1

To supplement the current images in Figures S9 and S10, which demonstrate the performance and SEM images of the hydrogel electrolyte after the liquid solution has aged for 0, 7, and 12 days, the author may consider including cycling performance and SEM images of the HA hydrogel electrolyte.

Response:

We agree with Reviewer #2 that including the cycling performance and SEM images of the Zn anode after being stored in the HA hydrogel electrolyte, will provide valuable insights into the understanding of corrosion reactions.

In response to this comment, we conducted additional electrochemical experiments and obtained microscopy data. Compared to the aged Zn//Zn cells in the liquid electrolyte (**Figure R16**), the cell using the HA gel electrolyte exhibits a more stable cycle performance even after 12 days' aging, although there is a slightly increased polarisation in the initial cycle (**Figure R17a**). The symmetric Zn//Zn cells can operate up to 300 hours without an increase in electrode polarisation (**Figure R17b**). Additionally, as demonstrated by SEM images in **Figure R18**, there is no ZHS by-product deposited on the immersed Zn anode, even after the 12-day aging process in the HA gel electrolyte. These findings provide further evidence of the effectiveness and durability of the HA hydrogel electrolyte in maintaining a stable chemical environment and enhancing the performance of the Zn anode.

Figure R16. Time-voltage profiles of aged Zn//Zn symmetric cells in liquid electrolyte at a current density of 2 mA cm^{-2} , with a total capacity of 2 mA h cm^{-2} : Profiles were recorded for (a) the initial 10 hours and (b) 300 hours.

Figure R17. Time-voltage profiles of aged Zn//Zn symmetric cells in the HA gel electrolyte at a current density of 2 mA cm^{-2} , with a total capacity of 2 mA h cm^{-2} : Profiles were recorded for (a) the initial 10 hours and (b) 300 hours.

Figure R18. SEM images of Zn anodes in aged Zn//Zn symmetric cells taken before and after 20 cycles at a current density of 2 mA cm^{-2} , with a total capacity of 2 mA h cm^{-2} in the HA gel electrolyte.

In response to address fully this comment, we have in our R-MS and R-SI, included following revision, namely:

1) R-SI, p. 31-32, included Figure S20 and Figure S21, namely:

Figure S20. Time-voltage profiles of Zn//Zn symmetric cells stored in the HA gel electrolyte at a current density of 2 mA cm^{-2} , with a total capacity of 2 mA h cm^{-2} : Profiles were recorded for (a) the initial 10 hours and (b) 300 hours.

Figure S21. SEM images of Zn anodes in aged Zn//Zn symmetric cells taken before and after 20 cycles at a current density of 2 mA cm^{-2} , with a total capacity of 2 mA h cm^{-2} in the HA gel electrolyte.

2) R-MS, p. 12, included in-text discussion, namely:

'In contrast, the cycling performance results demonstrate that the cycle life of Zn//Zn cells remains stable even after aging in the HA gel electrolytes for 12 days, despite of a slightly increased polarization in the initial cycle (Figure S20a). The symmetric Zn//Zn cells can operate up to 300 hours without an increase in electrode polarization (Figure S20b). Additionally, as demonstrated by SEM images in Figure S21, there is no ZHS deposited on the immersed Zn anode, even after the 12-day aging process in the HA gel electrolyte. These findings further evidence of the effectiveness and durability of the HA hydrogel electrolyte in maintaining a stable chemical environment and enhancing the performance of the Zn anode.'

Comment 2-2

It would be beneficial for readers' understanding if the author provided additional optical photos, including corresponding surface state diagrams of the HA hydrogel electrolyte on the zinc electrode or assembly diagrams of the HA hydrogel electrolyte in a button battery.

Response:

We agree with Reviewer #2 that additional optical photos should be included to show the surface state of the HA hydrogel electrolyte on the zinc electrode and to demonstrate the assembly of the HA hydrogel electrolyte in coin cell configurations.

In response to address fully this comment, we have, respectively, in our R-SI, p. 55 and p. 61, included Figure S42 and Figure S48, namely:

Figure S42. Schematic diagram of Zn-based coin cells using the HA gel electrolyte. Optical photo (right) shows the Zn/hydrogel electrolyte/Zn configuration.

Figure S48. Schematic diagram of flexible Zn/LMO full cells using the HA gel electrolyte. Optical photo (below) shows the cathode/hydrogel electrolyte/Zn configuration.

Comment 2-3

To further clarify the research, a simulation model diagram in Figure S9 of the Supporting Information could demonstrate the combination of the dominating cluster ($\text{Zn}(\text{H}_2\text{O})_6^{2+}$) with other clusters, such as $\text{ZnSO}_4(\text{H}_2\text{O})_5$ and $\text{ZnSO}_4(\text{H}_2\text{O})_4$, in the solvent structure.

Response:

We agree with Reviewer #2 that a simulation model diagram should be provided to demonstrate the combination of the dominating cluster ($\text{Zn}(\text{H}_2\text{O})_6^{2+}$) with other clusters.

In response to address fully this comment, we have in our R-SI, p. 37-38, included Figure S26 and Figure S27, namely:

Figure S26. Snapshots of the MD simulation model of the liquid electrolyte. Atom colours: Zn (grey); O (red); H (white); S (yellow).

Figure S27. Snapshots of the MD simulation model of the HA gel electrolyte. Atom colours: Zn (grey); O (red); H (white); S (yellow). C (indigo); N (blue).

Comment 2-4

The inclusion of optical photos of the HA hydrogel at different stretching degrees would provide a more detailed understanding of its properties.

Response

We agree with Reviewer #2 that optical photos of the HA hydrogel at different stretching degrees are needed to provide a more detailed understanding of the mechanical properties.

To assess mechanical properties of the HA gel electrolyte, we conducted additional experiments, including tests to determine the tensile strength and compressive strength. These results are shown in **Figure R19a** and **b**. The HA gel electrolyte exhibits a moderate elongation-at-break of 220% and a compressive strength of 0.18 MPa. The HA gel electrolyte can be stretched from 2 cm to 6 cm, representing a 200% increase in its original length, without any mechanical failure (**Figure R19c**). Our findings indicate that the HA gel electrolyte possesses good mechanical properties, which is a suitable candidate for flexible applications. These values are comparable or even exceed some of previously reported gel electrolytes, as summarised in **Table R4**.

Additionally, it was observed that, by reducing the water content in the HA gel electrolyte, the mechanical strength of the HA hydrogel can be further enhanced, which is promising for other applications that require high mechanical strength and stability (**Figure R20**). This enhancement may be accompanied by a minor trade-off, resulting in a slight reduction in ionic conductivity.

Figure R19. (a) Tensile and (b) compressive stress–strain curves of the HA gel electrolyte. (c) Optical photos showing the stretching process of the HA gel electrolyte.

Figure R20. (a) Tensile and (b) compressive stress–strain curves and (c) ionic conductivity of HA gel electrolytes with different water contents.

Table R4. The comparison of tensile strength and compressive strength with previously reported gel electrolytes. Abbreviations of gel electrolytes are as follows: Hyaluronic acid (HA), poly(acrylamide-co-[2-(methacryloyloxy)ethyl]dimethyl-(3-sulfopropyl)) (PASHE), poly(3-(1-vinyl-3-imidazolio)propanesulfonate) (PVIPS), polyacrylamide-poly (ethylene glycol) diacrylate-carboxymethyl cellulose (PMC), iota-carrageenan (IC), polyacrylamide (ZS/GL/AN), sorbitol-modified cellulose (Sor-Cel).

Gel electrolytes	Elongation-at-break (%)	Compressive strength (MPa)	Compressive strain (%)	Ref.
HA-6%	220	0.18	80%	This work
HA-15%	251	0.39	80%	This work
HA-30%	333	0.81	80%	This work
PASHE	315	0.05	60%	[3]
PVIPS	62	/	/	[5]
PMC	100	/	/	[7]
IC	128	/	/	[9]
ZS/GL/AN	350	/	/	[10]
Sor-Cel	303	0.085	50%	[11]

In response to address fully this comment, we have in our R-MS and R-SI, included following revision, namely:

- 1) R-SI, p. 12-14, included Figure S4, Figure S5 and Table S2, namely:

Figure S4. (a) Tensile and (b) compressive stress–strain curves of the HA gel electrolyte. (c) Optical photos showing the stretching process of the HA gel electrolyte.

Figure S5. (a) Tensile and (b) compressive stress–strain curves and (c) ionic conductivity of HA gel electrolytes with different water contents.

Table S2. The comparison of tensile strength and compressive strength with previously reported gel electrolytes. Abbreviations of gel electrolytes are as follows: Hyaluronic acid (HA), poly(acrylamide-co-[2-(methacryloyloxy)ethyl]dimethyl-(3-sulfopropyl)) (PASHE), poly(3-(1-vinyl-3-imidazolium)propanesulfonate) (PVIPS), polyacrylamide-poly(ethylene glycol) diacrylate-carboxymethyl cellulose (PMC), iota-carrageenan (IC), polyacrylamide (ZS/GL/AN), sorbitol-modified cellulose (Sor-Cel).

Gel electrolytes	Elongation-at-break (%)	Compressive strength (MPa)	Compressive strain (%)	Ref.
HA-6%	220	0.18	80%	This work
HA-15%	251	0.39	80%	This work
HA-30%	333	0.81	80%	This work
PASHE	315	0.05	60%	[14]
PVIPS	62	/	/	[16]
PMC	100	/	/	[18]
IC	128	/	/	[20]
ZS/GL/AN	350	/	/	[21]
Sor-Cel	303	0.085	50%	[22]

2) R-MS, p. 7, included in-text discussion, namely:

‘Figure S4a and b illustrates that the HA gel material displays a moderate elongation-at-break of 220% and a compressive strength of 0.18 MPa. The HA gel electrolyte can be stretched from 2 cm to 6 cm, representing a 200% increase in its original length, without any mechanical failure (Figure S4c). Additionally, by reducing the water content in the hydrogel, the mechanical strength of the HA gel electrolyte can be further enhanced for applications demanding high mechanical stability, at a minor cost of slightly decreased ionic conductivity (Figure S5). As summarised in Table S2, the mechanical strength

of the HA gel electrolyte is comparable or even exceed some of previously reported gel electrolytes, indicating its suitability as a candidate for flexible applications.'

Comment 2-5

I suggest that the graphs and corresponding units be checked more carefully; for example, paragraph 3 of the main text includes a superscript error whereby "mAh cm-3" should read as "mA h cm-2."

Response

We apologise for any confusion caused by the mistake.

For the text in paragraph 3, '*Among these systems, aqueous zinc metal batteries (AZMBs) are particularly attractive because of the utilisation of affordable and non-toxic Zn, which possess a favourable redox potential (-0.76 V vs. standard hydrogen electrode) and a high theoretical capacity (820 mA h g⁻¹ and 5855 mA h cm⁻³)*', the 5855 mA h cm⁻³ refers to the volumetric capacity of the Zn anode. We have taken additional steps to clarify the units and carefully reviewed and proofread the unit and formatting throughout the entire manuscript.

In response to address fully this comment, we have in our R-MS, p. 3, included following revision, namely:

'Among these systems, aqueous zinc metal batteries (AZMBs) are particularly attractive because of the utilisation of affordable and non-toxic Zn, which possess a favourable redox potential (-0.76 V vs. standard hydrogen electrode) and a high theoretical capacity (gravimetric capacity of 820 mA h g⁻¹ and volumetric capacity of 5855 mA h cm⁻³).'

Reviewer #3 (Remarks to the Author):

The development of batteries that possess both biocompatibility and long calendar life is crucial for the advancement of implantable devices in the future. This study by Li et al. reports a novel biocompatible hydrogel electrolyte based on hyaluronic acid (HA) to address the major challenges associated with aqueous zinc-based batteries for flexible and implantable devices. Those challenges include limited cycle life, side reactions, and biocompatibility. Compared to conventional liquid and other gel electrolytes, this HA gel electrolytes offer enhanced biosecurity with improved electrochemical performance, especially under a high DoD value of 80%, making it a promising alternative. The authors also conducted comprehensive investigations into the underlying mechanism of the HA gel electrolyte, including operando, spectral, and theoretical analyses. Therefore, this work serves as a valuable example and provides important theoretical insights for the progress of this field. In this case, it is recommended for publication in Nature Communications. However, some clarifications are necessary to strengthen the findings and conclusions.

Response

We thank Reviewer #3 for his /her comments and recommendation for publication.

Comment 3-1

The measurement of the ESW of electrolytes is commonly performed on Ti or GC electrodes due to their high HER overpotential. In this study, the authors employed Pt electrode as the working electrode, different from this common practice. The rationale behind this choice should be clarified.

Response:

We agree with the Reviewer #3 that the rationale for using the Pt electrode as the working electrode on the ESW measurement should be explained.

Generally, Pt, Ti, Glassy carbon (GC), etc. can be used as the working electrode for the three-electrode test. **Table R5** demonstrates that conventional electrodes, such as Ti and GC, exhibit high overpotentials for the HER, particularly at high current densities. As a result, the actual HER potentials on these electrodes in the liquid electrolyte (2M ZnSO₄, pH = 4.5) are lower than that of the Zn²⁺/Zn electrode potential (−0.76 V, **Figure R21**). For instance, the HER potential on the Ti electrode is −0.85 V, which indicates that the HER occurs after the Zn deposition.

As the deposited Zn exhibits, it becomes more difficult to distinguish between the HER and Zn deposition, because the increased overpotential of the Zn deposition leads to a stronger inhibition of the HER process during the LSV test.^[12] To mitigate the interference caused by the Zn deposition, we chose the Pt electrode as the working electrode due to its significantly smaller overpotential for the HER compared to other conventional working electrodes. The use of the Pt electrode as the working electrode was also reported in previous reports,^[3] which enables to determine and make comparisons of the HER potential occurred in two different electrolyte systems. In this case, the HER occurs before the Zn deposition (*e.g.*, −0.36 V_{HER} vs. −1.0 V_{Zn} for the liquid electrolyte), as shown in **Figure R22**. As a result, the finding shows that the HER potential is extended to −0.52 V in the HA gel electrolyte, indicating the superior reduction stability of our HA gel electrolyte.

Table R5. Comparison of HER potentials on various working electrodes. The HER overpotential is obtained from previous reports.^[13]

Electrode	HER overpotential (V)		
	$j=1000 \text{ mA cm}^{-2}$	$j=10 \text{ mA cm}^{-2}$	$j=1 \text{ mA cm}^{-2}$
Pt	0.44	0.39	0.09
Zn	1.24	0.75	0.72
Ti	0.82	0.6	/
Glassy carbon	1.13	/	/
Graphite	1.03	0.76	0.47
Stainless steel	/	0.42	/

Figure R21. The Zn^{2+}/Zn electrode potential and HER potentials on various working electrodes.

Figure R22. Magnified views of the regions outlined near (a) anodic and (b) cathodic extremes in (c) overall electrochemical stability window of liquid and HA gel electrolytes on non-active Pt electrodes.

In response to address fully this comment, we have in our R-MS and R-SI, included following revision, namely:

1) R-SI, p. 21-22, included Table S3 and Figure S12, namely:

Table S3. Comparison of HER potentials on various working electrodes. The HER overpotential is obtained from previous reports.^[13]

Electrode	HER overpotential (V)		
	$j=1000 \text{ mA cm}^{-2}$	$j=10 \text{ mA cm}^{-2}$	$j=1 \text{ mA cm}^{-2}$
Pt	0.44	0.39	0.09
Zn	1.24	0.75	0.72
Ti	0.82	0.6	/
Glassy carbon	1.13	/	/
Graphite	1.03	0.76	0.47
Stainless steel	/	0.42	/

Figure S12. The Zn^{2+}/Zn electrode potential and HER potentials on various working electrodes.

2) R-MS, p. 8, included in-text discussion, namely:

*'In addition, the electrochemical stability of both liquid and HA electrolytes was assessed by electrochemical stable windows (ESW), as demonstrated in **Figure 1c-e**. Note that the platinum (Pt) was utilised as the working electrode for its significantly smaller HER overpotential compared to other high-overpotential electrodes such as titanium and glassy carbon (**Table S3** and **Figure S12**). It ensures that the HER occurs prior to the Zn deposition, allowing for the distinct identification and separation of the two processes during the linear sweep voltammetry (LSV) test.'*

Comment 3-2

The HA electrolyte exhibits a mitigating effect on the corrosion of the Zn anode. To provide a comprehensive evaluation of this improvement, it is recommended to include the actual corrosion rate of Zn anodes in both electrolytes, instead of solely relying on the corrosion current as the sole metric.

Response:

We agree with the Reviewer #3 that the actual corrosion rate of Zn anodes in both electrolytes is needed to verify the mitigating effect of the HA gel electrolyte on the corrosion of the Zn anode.

We therefore measured the actual corrosion rate of metallic Zn by monitoring the potential variation of Zn@Ti electrode, which was prepared by plating a specifically quantitative amount of Zn on the Ti foil.

As is shown in **Figure R23**, the potential of the Zn@Ti electrode using the liquid electrolyte demonstrates a sudden increase only after 38 hours, indicating a complete consumption of the metallic Zn due to corrosion reactions.^[14] The corrosion rate ($Corr$) can be obtained by the equation: $Corr = \frac{m_{Zn}}{t}$, where m_{Zn} is the total mass of deposited Zn metal and t is the corrosion time corresponding to the complete consumption of Zn. Accordingly, the corrosion rate was calculated to be 0.0363 mg h^{-1} , which is significantly higher than the corrosion rate of 0.0068 mg h^{-1} observed with the HA gel electrolyte. This obvious difference in corrosion rates provides clear evidence of the excellent anti-corrosion capability of the HA gel electrolyte.

Figure R23. Voltage-time curves for Zn@Ti electrodes immersed in the liquid electrolyte and the H gel electrolyte. An amount of metallic Zn, 1.13 mAh, was deposited on Ti-foil to prepare the Zn@Ti electrode. Profiles were recorded for (a) the initial 5 hours and (b) 220 hours.

In response to address fully this comment, we have in our R-MS and R-SI, included following revision, namely:

- 1) R-SI, p. 23, included Figure S13, namely:

Figure S13. Voltage-time curves for Zn@Ti electrodes immersed in the liquid electrolyte and the HA gel electrolyte. An amount of metallic Zn, 1.13 mAh, was deposited on Ti-foil to prepare the Zn@Ti electrode. Profiles were recorded for (a) the initial 5 hours and (b) 220 hours.

2) R-MS, p. 10, included in-text discussion, namely:

*'To accurately quantify the corrosion rate of the Zn anode, we monitor the potential variation of a Zn@Ti electrode, where a predetermined amount of Zn was plated onto a Ti foil. As is shown in **Figure S13**, the potential of the Zn@Ti electrode using the liquid electrolyte demonstrates a sudden increase only after 38 hours, indicating a complete consumption of the metallic Zn due to parasitic side reactions and corrosions.^[17] The corrosion rate (Corr) can be obtained by the equation: $Corr = \frac{m_{Zn}}{t}$, where m_{Zn} is the total mass of deposited Zn metal and t is the corrosion time corresponding to the complete consumption of Zn. Accordingly, the corrosion rate was calculated to be 0.0363 mg h^{-1} , which is significantly higher than the corrosion rate of 0.0068 mg h^{-1} observed with the HA gel electrolyte. This obvious difference in corrosion rates provides clear evidence of the excellent anti-corrosion capability of the HA gel electrolyte.'*

Comment 3-3

The authors claimed that the coordination of Zn^{2+} and SO_4^{2-} is weaker in the HA gel electrolyte, based on findings from FTIR and DFT calculation. However, to strengthen this claim, it is recommended that the authors provide further evidence to substantiate it.

Response:

We agree with the Reviewer #3 that further evidence is required to support the weak coordination between Zn^{2+} and SO_4^{2-} .

We therefore determined the ion concentration and distribution near the Zn electrode by measuring the capacitance of electric double layer (EDL) on the Zn electrode. The EDL capacitance (C_{EDL}) was determined by the equation $C_{EDL} = \frac{\epsilon A}{d}$, where ϵ is the electrolyte dielectric constant, A is electrode surface area and d is the thickness of EDL. Considering both the liquid electrolyte and the HA gel electrolyte use water as the solvent and have the same Zn electrode size, ϵ and A can be considered roughly identical for comparison purpose. As a result, the C_{EDL} is mainly influenced by the thickness of the EDL, exhibiting a negative correlation.

As is shown in **Figure R24**, the EDL capacitance increases from 111 to 177 $\mu\text{F cm}^{-1}$ in the HA gel electrolytes, indicating a smaller d value in the presence of the gel electrolyte. Based on the classic stern model,^[15] this observation verifies that a greater concentration of Zn^{2+} ions but a reduced concentration of SO_4^{2-} ions accumulate on the surface of the Zn electrode in the HA gel electrolyte, attributed to the weak coordination between Zn^{2+} and SO_4^{2-} ions (**Figure R25**). By analysing the EDL capacitance, we understand the ion concentration and distribution near the Zn electrode, providing further substantiation of the weak coordination hypothesis between Zn^{2+} and SO_4^{2-} ions.

Figure R24. Cyclic voltammograms and current density-scanning rate fitting result of Zn//Zn symmetric cells using (a, c) the liquid electrolyte and (b, d) the HA gel electrolyte.

Figure R25. The comparison of the electric double layer: (a) between the Zn anode and the liquid electrolyte; (b) between the Zn anode and the HA gel electrolyte.

In response to address fully this comment, we have in our R-MS and R-SI, included following revision, namely:

1) R-SI, p. 41-42, included Figure S30 and Figure S31, namely:

Figure S30. Cyclic voltammograms and current density-scanning rate fitting result of Zn/Zn symmetric cells using (a, c) the liquid electrolyte and (b, d) the HA gel electrolyte.

Figure S31. The comparison of the electric double layer: (a) between the Zn anode and the liquid electrolyte; (b) between the Zn anode and the HA gel electrolyte.

2) R-MS, p. 16, included in-text discussion, namely:

*'Besides, the ion concentration and distribution near the Zn electrode was determined by measuring the capacitance of electric double layer (EDL) on the Zn electrode. The EDL capacitance (C_{EDL}) is governed by the equation $C_{EDL} = \frac{\epsilon A}{d}$, where ϵ is the electrolyte dielectric constant, A is electrode surface area and d is the thickness of EDL. Since both the liquid electrolyte and the HA gel electrolyte use water as the solvent and have the same Zn electrode size, ϵ and A can be considered roughly identical for comparison purpose. As a result, the C_{EDL} is mainly influenced by the thickness of the EDL, exhibiting a negative correlation. As is shown in **Figure S30**, the EDL capacitance increases from 111 to 177 $\mu\text{F cm}^{-1}$ in the HA gel electrolytes, indicating a smaller d value in the presence of the HA gel electrolyte. Based on the classic stern model,^[24] this observation verifies that a greater concentration of Zn^{2+} ions but a reduced concentration of SO_4^{2-} ions accumulate on the surface of the Zn electrode in the HA gel electrolyte, attributed to the weak coordination between Zn^{2+} and SO_4^{2-} ions (**Figure S31**). The red shift of SO_4^{2-} in FTIR spectra in **Figure S32** also verifies the weak chemical environment.^[25]*

Comment 3-4.

The authors should explain the significance of the increased Zn^{2+} transference number in the Zn plating/stripping process in the presence of the HA gel electrolyte.

Response:

We agree with the Reviewer #3 that the significance of the increased Zn^{2+} transference number in the Zn plating/stripping process should be explained.

The increased Zn^{2+} transference number plays a crucial role in improving the electrochemical performance of the zinc anode for two main reasons: 1) the enhanced mobility of Zn^{2+} ions helps alleviate the concentration gradient during the plating/stripping process. This, in turn, ensures a more uniform distribution of Zn^{2+} ions, leading to dendrite-suppressed Zn deposition. 2) The improved Zn^{2+} transference number also results in a decrease in the concentration of SO_4^{2-} anions near the Zn anode. The presence of high concentrations of SO_4^{2-} anions can induce undesired corrosion reactions, while the improved Zn^{2+} transference number contributes to mitigate the corrosion reactions induced by SO_4^{2-} anions.^[16]

In response to address fully this comment, we have in our R-MS, p. 17, included following revision, namely:

'The high Zn^{2+} transference number plays a crucial role in enhancing the electrochemical performance of the zinc anode through two key mechanisms: 1) allows for efficient transport of Zn^{2+} ions, alleviating the concentration gradient and ensuring a uniform Zn^{2+} ion distribution and thereby the dendrite-free Zn deposition. 2) decreases anion concentration near the Zn anode and mitigates SO_4^{2-} induced corrosion reactions.^[26]

Comment 3-5

The performance of Zn//LMO cells were evaluated under a demanding condition with a low N/P ratio (around 3.3). What is the cycle performance of the Zn//Zn symmetric cell under this condition?

Response:

We agree with the Reviewer #3 for the need to confirm the cycle performance of the Zn//Zn symmetric cell under a demanding condition with a low N/P ratio (around 3.3).

We therefore tested the cycle performance of Zn//Zn symmetrical cells under 33% depth of discharge (DoD), corresponding to a Zn utilisation rate of 33%. **Figure R26** illustrates the significant enhancement in the lifespan of the Zn anode when using the HA gel electrolyte. The HA gel electrolyte extends the lifespan from 140 hours to 650 hours, at a current density of 5 mA cm^{-2} and a capacity of 2.7 mA h cm^{-2} . This enhanced performance demonstrates the suitability of the HA gel electrolyte for practical applications where thin Zn foil ($<10 \text{ }\mu\text{m}$) is required to enhance the volumetric energy density and reduce the size of AZMBs for wearable or implanted devices.

Figure R26. Cycle performance of Zn//Zn symmetric cells with a high Zn utilisation rate of 33% DoD_{Zn} .

In response to address fully this comment, we have in our R-MS and R-SI, included following revision, namely:

- 1) R-SI, p. 53, included Figure S41, namely:

Figure S41. Cycle performance of Zn//Zn symmetric cells with a medium Zn utilisation rate of 33% DoD_{Zn} .

2) R-MS, p. 22, included in-text discussion, namely:

'To evaluate the potential of the HA gel electrolyte for practical applications, the cycle performance of Zn anode was further investigated under a medium DoD_{Zn} of 33% and a high DoD_{Zn} of 80%. Remarkably, in the HA gel electrolyte, the Zn//Zn symmetric cell delivers a stable cycling performance for 650 h under 33% DoD_{Zn} and even 250 h under 80% DoD_{Zn} (Figure 5h and S41).'

Comment 3-6

In addition to capacity and cycle life, the rate performance of full cells is also of great significance. To provide a comprehensive evaluation, it is recommended to include the rate performance data for the full cells.

Response:

We agree with the Reviewer #3 for the importance of the rate performance of full cells.

We therefore conducted further experiments on the rate capability of Zn//LMO and Zn//I₂ full cells, as shown in **Figure R27** and **Figure S28**. The results reveal that the HA gel electrolyte shown comparable rate performance with the liquid electrolyte using in Zn//LMO and Zn//I₂ full cells, which is mainly due to the similar ionic conductivity. Specifically, the Zn//LMO full cell maintains a specific capacity of 90.0 mA h g⁻¹ (75% of capacity at 0.5C), which is slightly lower than that of the liquid electrolyte (102.7 mA h g⁻¹, 84% of capacity at 0.5C). Furthermore, the Zn//I₂ cell using the HA gel electrolyte exhibits an impressive specific capacity of 150.0 mA h g⁻¹ even under an ultra-high current rate of 50C, only slightly lower than the specific capacity of 158.1 mA h g⁻¹ observed in the Zn//I₂ cell using the liquid electrolyte. These results demonstrate that the HA gel electrolyte show parallel rate capability with batteries utilising the liquid electrolyte, making it a promising choice for practical applications.

Figure R27. Rate capability of Zn//LMO cells in liquid and HA gel electrolytes.

Figure R28. Rate capability of Zn//I₂ cells in liquid and HA gel electrolytes.

In response to address fully this comment, we have in our R-MS and R-SI, included following revision, namely:

- 1) R-SI, p. 56, included Figure S43, namely:

Figure S43. Rate capability of Zn//LMO cells in liquid and HA gel electrolytes.

- 2) R-MS, p. 23, included in-text discussion, namely:

'The LMO cathode using the HA gel electrolyte maintains a comparable rate capability to that of ones using the liquid electrolyte, making it suitable for high-power applications (Figure S43).'

- 3) R-SI, p. 65, included Figure S52, namely:

Figure S52. Rate capability of Zn//I₂ cells in liquid and the HA gel electrolyte.

4) R-MS, p. 24, included in-text discussion, namely:

‘Because of the fast kinetics of the I₂ cathode, the cell using HA gel electrolyte delivers an impressive specific capacity of 150.0 mA h g⁻¹ at an ultra-high current rate of 50C (Figure S52), indicating its capability to provide a high power density.’

Comment 3-7

Some scratches can be observed on the Zn surface in Figure 2e. Did the authors conduct the pre-treatment on the Zn electrodes? If yes, it is important to provide experimental details regarding the preparation of the Zn anode.

Response:

Yes. All used Zn anode in our work were polished by a 3M softback (3M, U.S.A.) sanding sponge before use.

In response to address fully this comment, we have in our R-SI, p. 4, included in-text explanation, namely:

‘Before use, the Zn foil was polished by softback sanding sponge (3 M, U.S.A.) and then wiped with ethanol.’

Comment 3-8

A thorough revision of the entire manuscript is recommended to address typographical errors and inconsistencies. For example, the ZnSO₄(HA)(H₂O)₄ should be ZnSO₄(HA)(H₂O)₄⁻ in the legend of Figure S15.

Response:

We apologise for the typographical errors and inconsistencies. The entire manuscript has been carefully reviewed and proofread.

In response to address fully this comment, we have in our R-SI, p. 39, included Figure S28, namely:

Figure S28. The percentage of various Zn^{2+} solvation clusters in (a) the liquid and (b) the HA gel electrolytes obtained from MD simulations.

Reference:

- [1] a) D. Han, T. Sun, H. Du, Q. Wang, S. Zheng, T. Ma, Z. Tao, *Batteries Supercaps* **2022**, 5, 202200219; b) O. Fitz, C. Bischoff, M. Bauer, H. Gentischer, K. P. Birke, H. M. Henning, D. Biro, *ChemElectroChem* **2021**, 8, 3553.
- [2] Q. Liu, Z. Yu, R. Zhou, B. Zhang, *Adv. Funct. Mater.* **2022**, 33, 2210290.
- [3] W. Zhang, F. Guo, H. Mi, Z. S. Wu, C. Ji, C. Yang, J. Qiu, *Adv. Energy Mater.* **2022**, 12, 2202219.
- [4] C. Fu, Y. Wang, C. Lu, S. Zhou, Q. He, Y. Hu, M. Feng, Y. Wan, J. Lin, Y. Zhang, A. Pan, *Energy Stor. Mater.* **2022**, 51, 588.
- [5] Y. Hao, D. Feng, L. Hou, T. Li, Y. Jiao, P. Wu, *Adv. Sci.* **2022**, 9, 2104832.
- [6] Y. Liu, H. He, A. Gao, J. Ling, F. Yi, J. Hao, Q. Li, D. Shu, *Chem. Eng. J.* **2022**, 446, 137021.
- [7] P. Lin, J. Cong, J. Li, M. Zhang, P. Lai, J. Zeng, Y. Yang, J. Zhao, *Energy Stor. Mater.* **2022**, 49, 172.
- [8] J. Cong, X. Shen, Z. Wen, X. Wang, L. Peng, J. Zeng, J. Zhao, *Energy Stor. Mater.* **2021**, 35, 586.
- [9] Y. Tian, S. Chen, S. Ding, Q. Chen, J. Zhang, *Chem. Sci.* **2023**, 14, 331.
- [10] T. Wei, Y. Ren, Z. Li, X. Zhang, D. Ji, L. Hu, *Chem. Eng. J.* **2022**, 434, 134646.
- [11] Y. Quan, W. Zhou, T. Wu, M. Chen, X. Han, Q. Tian, J. Xu, J. Chen, *Chem. Eng. J.* **2022**, 446, 137056.
- [12] C. X. Zhao, L. Yu, J. N. Liu, J. Wang, N. Yao, X. Y. Li, X. Chen, B. Q. Li, Q. Zhang, *Angew. Chem. Int. Ed.* **2022**, 134, 202208042.
- [13] a) T. Zhang, J. Wu, J. Chen, Q. Pan, X. Wang, H. Zhong, R. Tao, J. Yan, Y. Hu, X. Ye, C. Chen, J. Chen, *ACS Appl. Mater. Interfaces* **2021**, 13, 24682; b) D. M. Heard, A. J. J. Lennox, *Angew. Chem. Int. Ed.* **2020**, 59, 18866; c) Q. Li, L. Han, Q. Luo, X. Liu, J. Yi, *Batteries Supercaps* **2022**, 5, 202100417; d) A. Hickling, F. W. Salt, *Trans. Faraday Soc.* **1940**, 36, 1226.
- [14] Y. Wang, Z. Wang, W. K. Pang, W. Lie, J. A. Yuwono, G. Liang, S. Liu, A. M. Angelo, J. Deng, Y. Fan, K. Davey, B. Li, Z. Guo, *Nat. Commun.* **2023**, 14, 2720.
- [15] X. Shi, J. Xie, F. Yang, F. Wang, D. Zheng, X. Cao, Y. Yu, Q. Liu, X. Lu, *Angew. Chem. Int. Ed.* **2022**, 61, 202214773.
- [16] a) K. Leng, G. Li, J. Guo, X. Zhang, A. Wang, X. Liu, J. Luo, *Advanced Functional Materials* **2020**, 30; b) W. Zhou, M. Chen, Q. Tian, J. Chen, X. Xu, C.-P. Wong, *Energy Storage Materials* **2022**, 44, 57.

*****End of Response Letter

REVIEWER COMMENTS

Reviewer #1 (Remarks to the Author):

The authors have revised the manuscript with some new data, which improves the quality. However, there are still some issues should be addressed before this paper can be accepted.

1. The authors claim that the thickness of the Zn anode should be 16 μm , yet the DOD can still keep at 80%. Usually, researchers calculate DOD by the thickness, for example 10 μm , 5.855 mA h cm^{-2} . Therefore, the DOD should be less than 70% for this work. So when should we employ thickness, and when for mass? Which one should be more standard and reliable?
2. What's the water ratio in the HA electrolyte? Does the ratio of water content influence the performance?
3. For S41, why it is asymmetrical during cycling?
4. What's the LMO loading for low N/P ratio of 3.3?
5. The XRD data of the Zn anode after Zn/LMO full battery long term cycling can be provided to further confirm the effect side reaction inhibition performance of HA electrolyte.

Reviewer #2 (Remarks to the Author):

The authors have addressed all issues in my previous report, and the paper is ready for publication.

Reviewer #3 (Remarks to the Author):

Previous questions have been addressed, it is now recommended for acceptance.

RESPONSE TO REVIEWS

Reviewer #1 (Remarks to the Author):

The authors have revised the manuscript with some new data, which improves the quality. However, there are still some issues should be addressed before this paper can be accepted.

Response:

We would like to thank the Reviewer #1 for his/ her insightful comments provided throughout the past two-round review process. These comments and suggestions significantly contributed to the enhancement of the quality of our research. In response, we have conducted additional experiments and provided explanations in detail to further substantiate our findings.

Comment 1-1

The authors claim that the thickness of the Zn anode should be 16 μm , yet the DOD can still keep at 80%. Usually, researchers calculate DOD by the thickness, for example 10 μm , 5.855 mA h cm^{-2} . Therefore, the DOD should be less than 70% for this work. So when should we employ thickness, and when for mass? Which one should be more standard and reliable?

Response:

We agree with the Reviewer #1 that researchers usually calculate the total capacity and the Depth of discharge (DoD) of the Zn anode by the thickness of the Zn. In this case, volume (V) and density (ρ) are used to calculate the mass (m) of the zinc foil ($m = \rho V$), followed by the calculation of the corresponding capacity of the Zn foil:

$$C_{\text{Zn}} = m_{\text{Zn}} \times 820 \text{ mA h g}^{-1}$$

However, it should be emphasised that this method assumes the theoretical density for the used Zn foil at 7.14 g cm^{-3} , which is identical to that of the pure Zn metal, and assumes that the surface of Zn foil to be both clean and smooth. Under this assumption, the measured thickness of the Zn foil is presumed to accurately represent the real thickness and to correspond proportionately with the mass of Zn foils.

However, the Zn foils used in our investigation have been polished, leading to an uneven surface (**Figure R1**). As a result, the measured thickness exceeds the actual value, introducing a disparity when we calculated the DoD by using the aforementioned approach. In addition, there is a difference between the practical density of the used Zn foil (16 μm) and the labelled value (10 μm) by the supplier, which may be caused by the intricacies during the production process. Considering the potential impact, we believe that using the aforementioned method to determine the DoD of the polished Zn anode in our manuscript is inappropriate.

In contrast, we determined the theoretical capacity of Zn anodes by using the actual mass of our polished Zn foils to circumvent these limitations. The reason behind this choice lies in the direct proportionality between mass and the theoretical capacity of the Zn anode ($C_{\text{Zn}} = m_{\text{Zn}} \times 820 \text{ mA h g}^{-1}$). In addition, the topmost oxidised layer can be removed by polishing, and its capacity contribution thereby can be neglected. Given these considerations, we find it more favorable to directly adopt the mass-based methodology to calculate the DoD of the Zn anode. This approach holds promise as a reliable method. Noted, the method of calculating capacity based on thickness is only applicable under specific circumstances, such as having an extraordinarily even Zn foil with a known practical density, criteria that are often demanding and time-intensive to confirm.

Figure R1. The scanning electron microscope (SEM) image of a polished Zn foil in our manuscript.

Comment 1-2

What's the water ratio in the HA electrolyte? Does the ratio of water content influence the performance?

Response:

In the preparation of HA gel electrolyte, the formulation was determined using a weight ratio of HA: H₂O: ZnSO₄, roughly amounting to 6:52.5:41.5 in weight. Notably, a small amount of water is anticipated to be evaporated under the preparation condition of 40 °C. To accurately understand the water ratio in the HA gel electrolyte after the preparation, we conducted thermogravimetric analysis (TGA) and the result can be found in the **Figure R2**. Specifically, the test involved gradually elevating the temperature from 50 °C to 150 °C at a heating rate of 10 °C min⁻¹, followed by maintaining a constant temperature of 150 °C to facilitate complete water evaporation but avoid the carbonization of HA molecules. The TGA curve indicates a slight decrease in the water content, bringing it down to 50.2% within the HA gel electrolyte. This finding indicates that the heating process has a minor impact on the overall water content in the HA gel electrolyte.

In order to explore the impact of water content on performance, we developed another two HA gel electrolyte with different water contents. Specifically, these variants included a 15% HA composition, which maintained a water content of 47.5%; and a 30% HA composition, accompanied by a water content of 39.0%. It is worth noting that as water content within the HA gel electrolyte decreases, a noticeable improvement in the mechanical strength of the HA hydrogel is observed (**Figure R3a** and **b**). However, this improvement is counterbalanced by a reduction in ionic conductivity, as illustrated in **Figure R3c**. This change in ionic conductivity, in turn, influences the operational lifespan of the Zn anode under conditions of high Zn utilization (80% DoD), which exhibits a reduced lifespan of 110 hours and 200 hours, respectively (**Figure R4**). As a result, after taking into account both ionic conductivity and mechanical strength, the optimal concentration of 6.0 wt.% is selected as the most suitable choice for the HA gel electrolyte formulation.

Figure R2. TGA curves of HA gel electrolytes with a temperature range increasing from 50 to 150 °C at a heating rate of 10 °C min⁻¹.

Figure R3. (a) Tensile and (b) compressive stress–strain curves and (c) ionic conductivity of HA gel electrolytes with different water contents.

Figure R4. Cycle performance of Zn//Zn symmetric cells using HA gel electrolyte with different water contents at a high Zn utilisation rate of 80% DoD_{Zn}.

In response to address fully this comment, we have in our R-SI, *p.* 2, included in-text explanation, namely:

'The HA gel electrolyte was prepared by mechanically mixing 6.0 wt.%, 15.0 wt.% and 30.0 wt.% HA powder with 2M ZnSO₄ aqueous solvent at 40 °C overnight. The water content of as-prepared gel electrolytes is approximately 52.5 wt.%, 47.5 wt.% and 39 wt.%, respectively.'

Comment 1-3

For S41, why it is asymmetrical during cycling?

Response:

At the beginning of charge-discharge process, there is no obvious asymmetrical cycling behavior observed (**Figure R5a and b**). However, after a long-term cycling, the asymmetrical voltage curves at the end of the cycling for Zn//Zn cells is observed, as is shown in **Figure R5c**. This can be attributed to the partial depletion of the Zn source within the Zn working electrode. Here, the Zn working electrode refers to the Zn metal at the positive electrode. In the charging process, Zn is gradually removed from the Zn working electrode. However, during the subsequent discharge process, only a portion of the Zn can retrace the path back to the Zn working electrode. This behavior is due to factors such as unavoidable corrosion reactions and/or irregular Zn deposition, especially at a relatively high DoD of 30% (**Figure R6**). Noted, though these two problems are significantly suppressed in the HA electrolyte, they cannot be completely eliminated. As a result, over extended cycles, the gradual loss of Zn accumulatively leads to the localised depletion of Zn within the system. This depletion has a detrimental impact on the stability of the electric network and reduces the accessibility of active sites. As a result, the energy barrier for the process of Zn stripping becomes more challenging to overcome, leading to the observed increase of the electrode potential during the charging process. This behaviours, characterised by asymmetrical voltage curves, is a commonly observed phenomenon during high DoD_{Zn} tests conducted on Zn//Zn cells.^[1]

Figure R5. Selected time-voltage profiles of Zn//Zn symmetric cells with a high Zn utilisation rate of 33% DoD_{Zn}. (a) Initial 5 hours, (b) 138–143 hours, and (c) 645–650 hours.

Figure R6. Illustration of the change of Zn electrodes during long-term cycling with a high DoD_{Zn} .

In response to address fully this comment, we have in our R-SI, *p.* 53, included Figure S41 and in-text explanation, namely:

Figure S41. (a) Cycle performance of Zn//Zn symmetric cells with a relatively high Zn utilisation rate of 33% DoD_{Zn} . (b) Illustration of the change of Zn electrodes during long-term cycling with a high DoD_{Zn} .

The asymmetrical voltage curves at the end of the cycling (Figure S41a) of Zn//Zn cells is attributed to the partial depletion of the Zn source of the Zn working electrode. Here, the working electrode is defined as the Zn metal at the positive electrode. In the charging process, Zn is gradually removed from the Zn working electrode. However, during the subsequent discharge process, only a portion of the Zn can retrace the path back to the Zn working electrode. This behavior is due to factors such as unavoidable corrosion reactions and/or irregular Zn deposition, especially at a relatively high DoD of 30% (Figure 41b). Noted, though these two problems are significantly suppressed in the HA electrolyte, they cannot be completely eliminated. As a result, over extended cycles, the gradual loss of Zn accumulatively leads to the localised depletion of Zn within the system. This depletion has a detrimental impact on the stability of the electric network and reduces the accessibility of active sites. As a result, the energy barrier for the process of Zn stripping

becomes more challenging to overcome, leading to the observed increase of the electrode potential during the charging process.'

Comment 1-4

What's the LMO loading for low N/P ratio of 3.3?

Response:

The mass loading is about 11.6 mg cm^{-2} for the LMO cathode used in the Zn//LMO full cells with a low N/P ratio of 3.3. We calculated the mass loading of LMO based on the practical specific capacity of 120 mA h g^{-1} and the electrode area of 1.131 cm^2 (with a diameter of 12 mm). In this case, the total capacity of the LMO cathode is calculated to be 1.57 mA h. To maintain a consistent N/P ratio of 3.3, we used a thin Zn anode (a thickness of $16 \text{ }\mu\text{m}$ with an areal capacity of $8.13 \text{ mA h cm}^{-2}$ in total) with a diameter of 9 mm, which has an actual capacity of 5.17 mA h. Therefore, the N/P ratio is calculated to be 3.3.

In response to address fully this comment, we have, respectively, in our R-SI, *p.* 2 and *p.* 4, included in-text explanation, namely:

'The areal mass loadings of the LMO cathode and I₂ cathode were controlled at 11–12 mg cm⁻² and 2.0–2.5 mg cm⁻², respectively.'

'The cathodes and Zn anodes were cut as disk-shaped electrodes with a diameter of 12 mm for the coin cell assembly if not mentioned, while Zn anodes with a smaller diameter of 9 mm were used in the Zn//LMO full cells with a low N/P ratio of 3.3.'

Comment 1-5

The XRD data of the Zn anode after Zn/LMO full battery long term cycling can be provided to further confirm the effect side reaction inhibition performance of HA electrolyte.

Response:

We agree with Reviewer #1 that the XRD data of the Zn anode in the Zn/LMO full battery after long-term cycling is needed, which helps to further confirm the side reaction inhibition role of the HA electrolyte.

As shown in **Figure R7**, after cycling in Zn//LMO full cells for 100 cycles, the Zn anode using the liquid electrode exhibits a high amount of $\text{Zn}_4\text{SO}_4(\text{OH})_6 \cdot 3\text{H}_2\text{O}$ (ZHS) by-product. In contrast, only a minor amount of ZHS is observed on the Zn anode using the HA gel electrolyte even after 500 cycles, further demonstrating the exceptional corrosion inhibiting capability of the gel electrolyte.

Figure R7. XRD spectra of cycled Zn metals in Zn//LMO full cells at 3C using liquid and HA gel electrolytes.

In response to address fully this comment, we have in our R-MS and R-SI, included following revision, namely:

- 1) R-SI, p. 59, included Figure S46, namely:

Figure S46. XRD spectra of cycled Zn metals in Zn//LMO full cells at 3C using liquid and HA gel electrolytes.

- 2) R-MS, p. 23, included in-text discussion, namely:

'More importantly, the hybrid AZMBs exhibits noticeable improvement in cycle stability, with a capacity retention of 82% after 1000 cycles (Figure S44), in contrast to those using the liquid electrolyte which fails after only 133 cycles due to short circuits (Figure S45). In addition, the significantly lower presence of ZHS by-products on the cycled Zn anode in the gel electrolyte further verifies the exceptional corrosion inhibiting capability of the HA gel electrolyte (Figure S46).'

Reviewer #2 (Remarks to the Author):

The authors have addressed all issues in my previous report, and the paper is ready for publication.

Response

We thank Reviewer #2 for the positive comments and recommendation for publication in Nature Communications.

Reviewer #3 (Remarks to the Author):

Previous questions have been addressed, it is now recommended for acceptance.

Response

We thank Reviewer #3 for the positive comments and recommendation for publication in Nature Communications.

Reference:

- [1] a) J. Li, Q. Lin, Z. Zheng, L. Cao, W. Lv, Y. Chen, *ACS Appl. Mater. Interfaces* **2022**, 14, 12323; b) P. Cao, X. Zhou, A. Wei, Q. Meng, H. Ye, W. Liu, J. Tang, J. Yang, *Adv. Funct. Mater.* **2021**, 31, 2100398; c) J. Wang, H. Qiu, Q. Zhang, X. Ge, J. Zhao, J. Wang, Y. Ma, C. Fan, X. Wang, Z. Chen, G. Li, G. Cui, *Energy Stor. Mater.* **2023**, 58, 9; d) Z. Zhao, W. Yu, W. Shang, Y. He, Y. Ma, Z. Zhang, P. Tan, *J. Power Sources* **2022**, 543, 231844; e) Z. Xu, N. Zhang, X. Wang, *Nano Energy* **2022**, 102, 107724; f) M. Kwon, J. Lee, S. Ko, G. Lim, S.-H. Yu, J. Hong, M. Lee, *Energy Environ. Sci.* **2022**, 15, 2889.

*****End of Response Letter

REVIEWERS' COMMENTS

Reviewer #1 (Remarks to the Author):

I would like to suggest the acceptance of this version of revised manuscript